

# Privacy-preserving solution for vehicle parking services complying with EU legislation

Petr Dzurenda[1], Florian Jacques[2], Manon Knockaert[2], Maryline Laurent[3], Lukas Malina[1], Raimundas Matulevicius[4], Qiang Tang[5] and Aimilia Tasidou[3]

[1] Department of Telecommunications, Brno University of Technology, Brno, Czech Republic
[2] University of Namur, Namur, Belgium
[3] Samovar, Télécom SudParis, Institut Polytechnique de Paris, Paris, France
[4] Institute of Computer Science, University of Tartu, Tartu, Estonia
[5] IT for Innovative Services Department, Luxembourg Institute of Science and Technology, Luxembourg

## ABSTRACT

Today, many modern cities adopt online smart parking services as best practices. Citizens can easily access these services using their smartphones or the infotainment panels in their cars. These services' primary objective is to give drivers the ability to quickly identify free parking slots, which should reduce parking time, save fuel, and relieve traffic in urban areas. However, the privacy offered by these services should be comparable to that of the standard paper-based parking solutions offered by parking ticket machines. On the other hand, a privacy-preserving smart parking service's design may raise a number of issues, including how to prevent double or multiple uses of parking tickets, how to prevent user tracking and profiling, how to revoke malicious users, how to handle data statistics without violating users' privacy, and how to comply with regulations like the General Data Protection Regulation (GDPR). In this article, we present multidisciplinary research on a comprehensive vehicle parking system that protects users' privacy. The research includes a range of topics, from the examination of regulatory compliance to the design of privacy-preserving parking registration and vehicle parking services to the implementation of privacy-preserving parking data processing features for data analysts. We provide a security analysis of our concept as well as several experimental results.

# INTRODUCTION

In future smart cities, smart parking solutions will be more and more integrated with city services and used by numerous citizens via their smartphones or infotainment panels in their vehicles. The main goal of smart parking services is to provide drivers the efficient detection of vacant slots that should shorten the time during parking, save fuel and decrease congestion in cities. Parking in city streets and parking lots is usually for specific fees according to different zones, periods, and daily times. Therefore, parking services usually have to collect these fees from users. Using parking lot terminals with tollgates

Corresponding author
Raimundas Matulevicius, rma@ut.ee

should help with a payment collection, and only users who paid for the service should get access to the parking lots. Nevertheless, using prepayment and intelligent detection of free slots via mobile applications causes that users have to interact remotely with a smart parking system in advance. These systems should ideally provide a similar level of privacy as the traditional paper-based parking solutions with parking ticket machines. The design of a privacy-preserving smart parking service may open several issues such as how to prevent double/multiple spending of parking tickets, how to prevent user tracking and linking, how to revoke malicious users, how to handle data statistics without privacy breaches and how to be compliant with regulations such as General Data Protection Regulation (GDPR).

In this article, we proposed a novel privacy-preserving solution for vehicle parking services which is complying with European Union (EU) legislation, especially with privacy and security requirements defined by current regulations and directives. The system protects users' privacy and their digital identities. Furthermore, it also allows third parties such as research institutions to run statistical analyses on parking data. This analysis can be done without impacting the privacy of both, *i.e.,* users (no personal data or linkable information about users are disclosed) and analysts (no information about what they are searching for is revealed). To do so, we had to answer three main research questions (RQ):

- **RQ1:** What are the legal instruments, issues, and requirements for the deployment of such a system?
- **RQ2:** How to build a privacy-preserving system which meets the requirements from RQ1? Which Privacy-Enhancing Technology (PET) can be used in order to protect users' privacy during using the system, *i.e.,* reservation of parking slots and parking vehicle actions?
- **RQ3:** How to allow third parties to perform statistical analyses on the parking transaction data, in a privacy preserving way? Which PET can be used to support this task?

The article is organized as follows. The 'Related Work' section analyzes the recent research on security and privacy in smart cities with a focus on parking service applications. 'Parking Scenario Description' introduces a high-level architecture description of our parking system, security and privacy requirements. 'Legal Issues' presents the different legal instruments relevant for the deployment of vehicle parking systems. 'Cryptographic Preliminaries' outlines the used notation needed to understand the cryptographic design of our parking system. 'Privacy-Preserving Parking Solution' introduces our privacy-preserving parking system, its security analysis, and experiment results. 'Privacy Preserving Data Processing for Statistics Analysis' presents our solution for privacy-preserving parking data processing, its security analysis, and experimental results. In 'Concluding remarks', we conclude this work.

## RELATED WORK

In many existing works, smart parking services are usually considered as the part of smart cities or intelligent infrastructures. There are several works that deal with general security and privacy issues in smart cities and deal partially with parking services, such as *Martínez-Ballesté, Pérez-Martínez & Solanas (2013)*; *Al-Turjman & Malekloo (2019)*; *Al-Turjman,*

*Zahmatkesh & Shahroze (2019)* and *Navaroj & Julie (2021)*. Further, privacy-preserving smart parking solutions and parking related problems in cities have been introduced in recent works such as *Garra, Martínez & Sebé (2016)*; *Chatzigiannakis, Vitaletti & Pyrgelis (2016)*; *Huang et al. (2018)*; *Zhu et al. (2018)*; *Borges & Sebé (2019)*; *Al Amiri et al. (2019)*; *Fang et al. (2021)*; *Dzurenda et al. (2021)* and *Khalid et al. (2021)*.

For example, *Garra, Martínez & Sebé (2016)* proposed a practical privacy-preserving pay-by-phone parking system based on periodical e-coin micro-payments for short intervals. The proposal deploys Hash-Based Message Authentication Codes (HMAC), RSA signatures, Chaum's blinded signatures based on RSA introduced in *Chaum (1983)* and Elliptic Curve Digital Signature Algorithm (ECDSA) signatures. The drawback of the proposal can be technical issues such as lack of coverage, low battery, etc.

In *Borges & Sebé (2019)*, the authors claimed that it has solved these technical disadvantages in their proposal of a privacy-preserving pay-by-mobile parking system. Their e-coin based proposal offered the same privacy as the traditional paper-based approach. Users' privacy is preserved without requiring a trusted party. The proposal deploys the Chaum's blinded signatures based on RSA and Digital Signature Algorithm (DSA). Later, *Borges & Sebé (2021)* presented an upgraded and more efficient solution than in *Borges & Sebé (2019)*. Nevertheless, both solutions digitally collect also car plate numbers (licenses) by parking officers.

*Chatzigiannakis, Vitaletti & Pyrgelis (2016)* investigated privacy-preserving smart parking systems using the IoT platform. They adopted Elliptic Curve Cryptography (ECC) as an attractive alternative to RSA-based solutions. They showed how to deploy zero-knowledge proofs (ZKP) using ECC that should preserve users' privacy. Moreover, they created a real-world outdoor IoT testbed and analyzed the execution time on various IoT platforms. Their work did not provide a tailored proposal but offered interesting practical results.

*Huang et al. (2018)* presented a secure and privacy-preserving reservation/parking solution for automated valet parking systems without a trusted third party. Their solution is based on zero-knowledge proofs proposed by *Fiat & Shamir (1986)*, geo-indistinguishable mechanism published in *Andrés et al. (2013)*, proxy re-signatures designed by *Libert & Vergnaud (2008)*, and bloom filter data structure. Their parking reservation costs almost 3 s due to deploying the heavy cryptographic operations. *Zhu et al. (2018)* focused on smart parking in cities and presented the anonymous smart-parking and payment scheme in vehicular networks. Their solution is based on the Pointcheval-Sanders randomizable signature designed by *Pointcheval & Sanders (2016)* and using a trusted authority. For generating a parking query, one driver has to compute several exponentiations, multiplications, additions, and hash.

*Al Amiri et al. (2019)* presented a privacy-preserving smart parking system using blockchain and private information retrieval. A shared ledger should increase security, transparency, and availability. The system preserves drivers' location privacy by using the private information retrieval of parking offers from the blockchain nodes and deploying short randomizable signatures proposed in *Pointcheval & Sanders (2016)* allow drivers to anonymously reserve available parking slots. The reservation time is around 1 ms at 1.2 GHz

processor with 160-bits MNT curve and SHA-2. Similarly, *Fang et al. (2021)* presented a blockchain-based privacy-preserving valet parking protocol. The solution is based on a new variant of Pointcheval-Sanders group signature, and it is secure in the random oracle model. Blockchain-based privacy-preserving decentralized parking recommendation solutions has been also proposed by *Li et al. (2021)*. Their solution employs a private blockchain, a bulletin board, a re-randomized homomorphic encryption scheme, zero-knowledge protocols and oblivious pseudorandom functions. Recently, *Dzurenda et al. (2021)* have proposed the privacy-preserving online parking system based on blockchain and smart contracts. The system deploys provable secure cryptographic primitives such as revocable anonymous credential proposed in *Hajn et al. (2021)* and partially blinded signature proposed in *Abe & Okamoto (2000)*. The system provides a full set of privacy-enhancing features such as user anonymity, untraceability, and unlinkability. Furthermore, the authors involve blockchain and smart contracts technologies in the payment and verification phases to make the system more transparent, decentralized, and resistant against cyberattacks.

The complex taxonomy of smart parking and autonomous valet parking solutions has been presented in the recent survey by *Khalid et al. (2021)*. This survey studies many aspects of parking solutions, where security and data privacy processing have been detected as ones from challenges and future directions.

Few related works have also studied legal challenges and regulations in smart cities and parking services. For example, *Weber & Podnar Žarko (2019)* provides a regulatory view on smart city services where smart parking systems are integrated, and *Losavio et al. (2018)* deals with legal challenges in smart cities. Nevertheless, a detailed study focusing on the regulation requirements of smart parking systems is still missing.

In this article, we focus on a complex spectrum of problems in privacy-preserving smart parking including legal and technical perspectives in order to cover various layers (authentication, secure communication, data processing, and other aspects). Our multidisciplinary work presents a comprehensive privacy-preserving proposal for parking services that covers privacy-preserving parking requests, privacy-preserving data statistics, regulation compliance, and other privacy issues related to communication and system settings.

# PARKING SCENARIO DESCRIPTION

In this section, we present a high-level system architecture, and we define the system entities, the parking scenario phases, and the privacy and security requirements.

## System architecture

Three types of entities interact in our privacy-preserving vehicle parking system:

- **Parking Service Provider** (PSP): The PSP generates cryptographic parameters and keys. It also registers new users and revokes/identifies the malicious ones. Furthermore, the PSP mediates communication between users and the PLT and enrolls new PLTs in the system. The communication with the PSP takes place fully via an Internet connection. We assume that the PSP is a semi-trusted party which honestly runs the algorithms but could be curious.

- **Parking Lot Terminal** (PLT): The PLT represents the system controlling access to the specific parking lot. It is responsible for issuing the parking permits to users and verifying the presented parking permits by users. The communication with the PSP takes place via an Internet connection during the parking permit issue phase (reservation parking in the parking lot) and via Bluetooth connection during the parking permit verification phase (accessing the parking lot).
- **User Device**: The user is represented by its device, typically a smartphone. These devices allow storing users' parking permits issued by the PLT through the PSP and presenting these permits to the PLT when users access the parking lot. Furthermore, this device holds system parameters, generates and stores user cryptographic keys, communicates with PSP via Internet connection (*i.e.,* Wi-Fi, Long Term Evolution (LTE)) and with PLT via the Bluetooth Low Energy (BLE) interface.

The privacy-preserving parking system with all involved entities and protocols is depicted in Fig. 1. The proposal also involves a trusted third party—IDentity Provider (IDP) that manages user identity and associated identity attributes.

## Trust assumptions

We assume that communication between all communication parties is secured. In particular, the communication between users and the PSP and the communication between the PSP and PLTs is secured by Transport Layer Security (TLS) protocol. The whole system is based on a trust chain, *i.e.,* we expect the existence of Public Key Infrastructure (PKI) and trusted certification authorities. Besides the privacy-enhancing protocols used in our parking system in each scenario phase described in the 'Detailed Description of Our Algorithms' section, we need to consider also other privacy issues which can impact users privacy:

- **Anonymous Payment Methods**: The payment to PLT can be done privately by deploying the improved e-payment 3D-Secure protocol (*Plateaux et al., 2013*) or by using popular wallets on mobile devices that support privacy-preserving cryptocurrencies, *i.e.,* Monero (*Noether, 2015*), Zcash (*Kappos et al., 2018*), and DASH (*Duffield & Diaz, 2015*). The security and privacy of popular Android wallets have been studied in *Biryukov & Tikhomirov (2019)*.
- **Anonymous Communications in Wide Area Network (WAN)**: Privacy-preserving communication in WAN can be achieved by mature onion routing protocols and techniques such as ToR (*Dingledine, Mathewson & Syverson, 2004*). Then, users are able to privately communicate with PSP via Internet during their registration and issuing parking permit. On one hand, users' source addresses and actual locations are hidden to PSP and observers because the ToR protocol applies at least three randomly-selected servers (onion routers) as relays and encrypts the communication (creating the onion layers). On the other hand, communication via ToR can cause delays due to encryption operations and using more hops.
- **Anonymous Communications in Personal Area Network (PAN)**: For PAN, one typical technology is Bluetooth Low Energy (BLE). By design, BLE provides a reasonable level

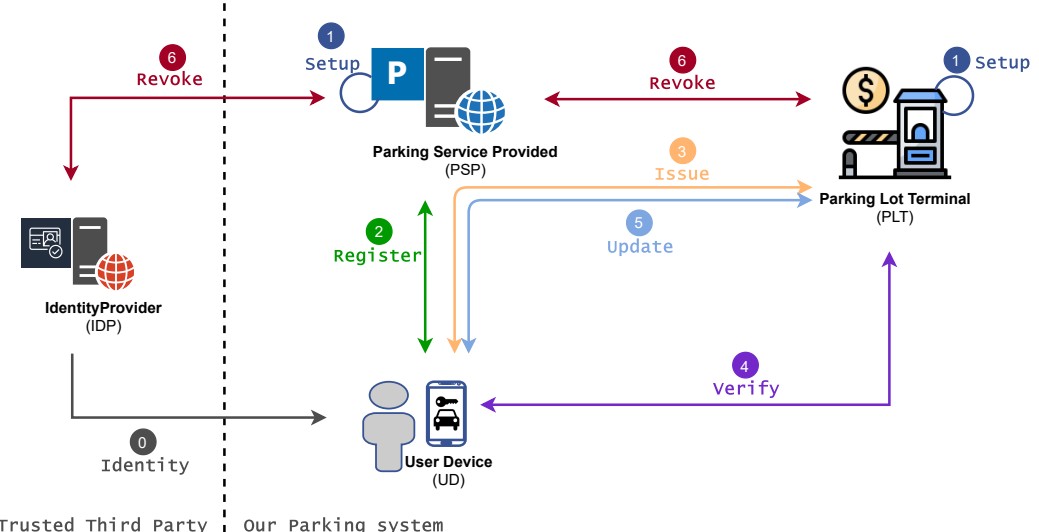

**Figure 1** Privacy-preserving parking system.

of privacy protection with features like address randomization (*Cäsar et al., 2022*) and it has been widely used in contact tracing for COVID-19, *e.g.*, *Tang (2022)*. Therefore, we can assume that BLE provides a sufficient level of anonymity/privacy guarantee in our application.

- **Surveillance Minimization**: Surveillance security systems with cameras are usually deployed in parking lots and garages in order to increase security against various physical attacks, vandalism, and thefts. Moreover, some solutions are based on using Automatic Number Plate Recognition (ANPR) or Licence Plate Recognition (LPR) to detect concrete vehicles that prepaid a service. Nevertheless, these camera systems could conflict with users' privacy and GDPR. Thus, it is necessary to use records and basic functionality of ANPR only for security purposes and not to store records for longer periods or non-permitted tracking.

## Scenario phases

Our parking scenario consists of the following phases:

1. **Register user phase**: The digital identity of the user is created in this phase, as illustrated in Fig. 2. First, the users download the mobile application of the parking system, *e.g.*, using Google Play (see, *1.1. Download mobile parking application*). Second, the users use the application to create their; own digital identity in the parking system. To do so, we suggest involving a trusted third party that will manage user identity and associated identity attributes (see, *1.2. Create digital identity (through trusted third party)*). This party is called IDentity Provider (IDP). Thanks to using the IDP, we do not need to store sensitive user data, such as name, surname, address, age, gender. Otherwise, these data can directly identify the user and can be a target of cyberattacks. Therefore, we suggest to deploy one from these following methods to create a digital identity in the parking system:

- **Payment card binding**: The users have to add their bank cards to the parking application. The PSP does a pre-authorization charge to make sure the payment card used by the user is valid. If so, the PSP will create the digital identity of the user. The digital identity is represented by the payment card number provided by the user. The PSP learns no more information about the user. If the user commits fraud, the PSP will query disclosing the user identity to the bank that issued the payment card.

- **Mobile number binding**: The users have to add their phone number to the parking application. The PSP sends an authorization code to this phone number to make sure the phone number provided by the user is valid. If so, the PSP will create the digital identity of the user. The digital identity is represented by the phone number provided by the user. The PSP learns no more information about the user. If the user commits fraud, the PSP will query disclosing the user identity to the mobile operator. In this case, it is necessary to have a registered telephone number, such as in some European Union (EU) countries. For example, all SIM cards in Spain need to be registered by law.

- **Electronic identification (eID) binding**: The user has to use trusted Identity Provider (IDP) supported by the PSP and according to the EU electronic identification and trust services (eIDAS) Regulation. For example, in the Czech Republic, we can find the eObanka application. The PSP learns no more information about the user. If the user commits fraud, the PSP will query disclosing the user identity to the organization delivering public digital services in an EU member state.

When the digital identity of the user is created, the PSP runs the `Register` algorithm (see, *1.3. Register user*). In particular, the PSP generates the user access credential $\Lambda$ and user secret key $sk_U$. To do so, the PSP will use group signature (*Hajny et al., 2018*). The credential and the secret key are sent to the user's device (see, *1.4. Send user credentials*) where they are securely stored (see, *1.5. Store credentials and secret key*). Furthermore, the secret key is also stored in the PSP Revocation Database (RD) (see, *1.6. Store secret key*). The PSP can use this database to revoke or identify malicious users. The user revocation is possible only in collaboration with the PLT. The user identification requires also the involvement of IDP.

2. **Issue parking permit phase**: The parking reservation is made through the PSP. The PSP acts as a gateway between the user and the PLT, as illustrated in Fig. 3. The PSP does not interfere with the `Issue` algorithm. It only forwards the communication between communicating parties. The `Issue` algorithm is run between the user device and the PLT. First, the user sends a parking request to the PLT (see, *2.1. Send parking request*). Basically, this information is where, when, and for how long the user wants to park. No sensitive, personal, or other linkable data are provided. This information is sent in a clear way, and therefore both the PSP and the PLT know them. Furthermore, the reservation request also includes the user access credential $\Lambda$ issued by the PSP. This credential is blinded, and therefore, the PSP nor PLT can learn it in this phase. Additionally, the access credential $\Lambda$ is randomized with a session credential key $sk_{Cred}$. This key is generated by the user for each new reservation phase, and therefore, it

differs for all user's parking permits. Second, after the payment for the parking is done (see, *2.3. Perform payment*), the PLT generates parking permit ID (see, *2.5. Generate parking permit*) and computes a partially blind signature (*Abe & Okamoto, 2000*) on parking request data (both, clear and blind information) and sends it to the user (see, *2.6. Send parking permit*). The user uses the partially blind signature from the PLT to reconstruct the parking permit *CRED*. The PSP and the PLT do not see the whole parking permit. They see only its public data, *i.e.,* parking permit ID (PPID), parking location (PLTid), parking time (time_duration) and information about parking time extension (EPT).

3. **Park vehicle phase**: The user accesses the parking lot in this phase as illustrated in Fig. 4. To get access, the user must authenticate to the PLT first (see, *3.1. Authenticate* and *3.2. Confirm*). During the parking vehicle phase, the user communicates directly with the PLT, for example, via a Bluetooth communication interface. First, the user sends the parking permit to the PLT (see, *3.3. Send parking permit*). The PLT checks the parking permit data and verifies the signature on the permit using PLT's public key $pk_{PLT}$ (see, *3.4. Verify signature*). If the parking permit is valid, the users must prove that the parking permit belongs to them; (see, *3.5. Check user authentication proof*), *i.e.,* the permit includes the access credential $\Lambda$ issued by the PSP and randomized by the user with the credential key $sk_{Cred}$. To do so, the user and the PLT run the Verify algorithm. The PLT checks the user's authentication proof using PSP's public key $pk_{PSP}$. If the proof is valid, the user is allowed to access the parking lot and the barrier is opened (see, *3.6. Allow vehicle to enter* and *Enter PLT and park vehicle*). The parking permit includes user access credential $\Lambda$ which can be used for linking the parking permit to the real identity of the user. However, this access credential is randomized with different credential keys in all issued user's parking permits. Therefore, the PLT cannot link two different parking permits to the one user, and therefore, the user access parking lot anonymously and unlinkably. This prevents the possibility of profiling and tracking users across the system. The PLT is not able to get any information on how often users park their vehicles in the parking lot.

4. **Extend parking time phase**: The users can extend their parking time period using the Update algorithm as illustrated in Fig. 5. Users do not need to reveal any personal data to extend the parking time. The main assumption is that the PLT already has the user's parking permit, *i.e.,* the user parked the vehicle in the parking lot. First, the user sends the extension parking time request to the PSP (see, *4.1. Send the extension parking time request*). This request includes PPID and PLTid information. Thanks to PLTid the PSP finds the relevant PLT (see, *4.2 Transfer the extension parking time request*). Because of the PPID, the PLT finds the relevant parking permit (see, *4.3 Find relevant parking permit*). If the extension parking time is possible (see, *4.4. Check if extension is possible*), then the user and the PLT run the Verify algorithm in order to authenticate and authorize the user. If the user is authenticated, then the user and the PLT run the Issue algorithm with a new extended time period (see, *4.7. Issue parking permit* and Fig. 3). The Issue algorithm is run after the payment for the extended parking time is made.

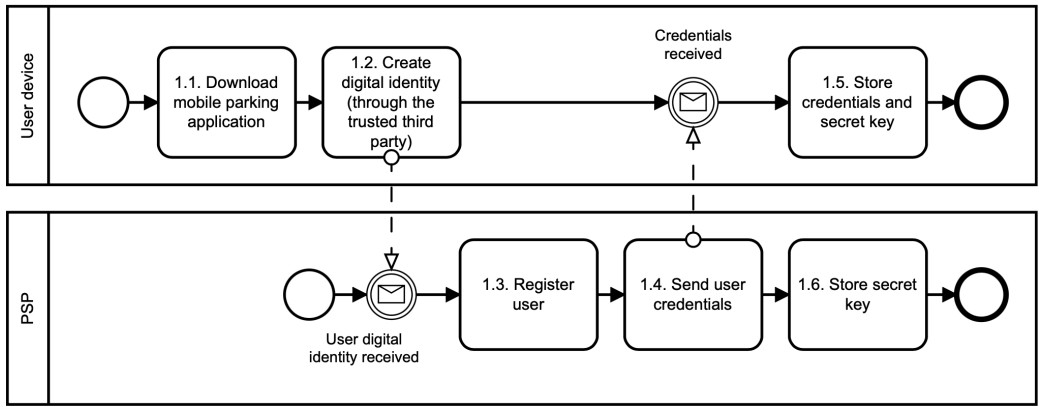

**Figure 2** Register user.

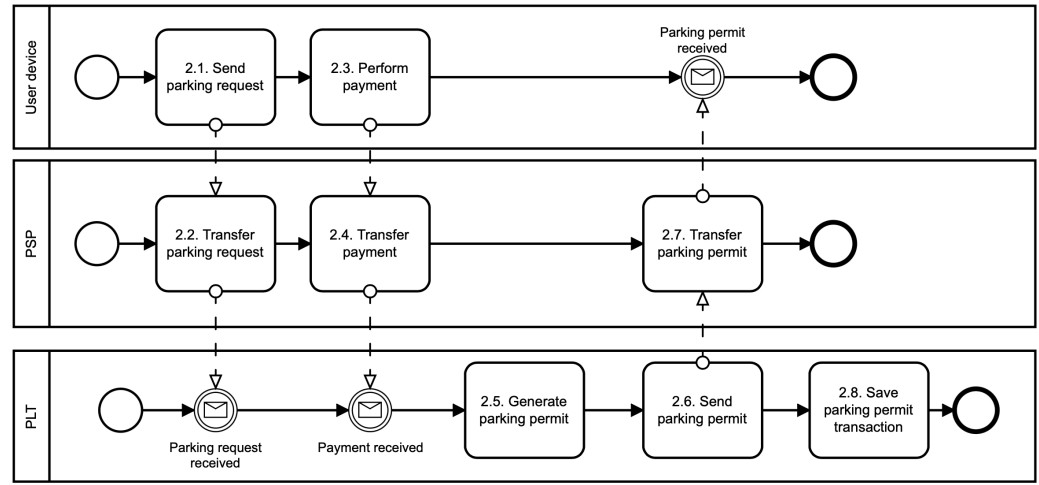

**Figure 3** Issue parking permit.

## Privacy and security requirements

This section introduces general security and privacy requirements on the parking reservation system. In particular, we have the following system security requirements:

- **Authentication**: Parking permits from PSP should be granted only to valid non-revoked users who use them in the parking phase. The users should stay in anonymity but should prove that they hold valid parking permits (based on reservation) to PLT when the user arrives at a parking lot.
- **Data confidentiality**: All sensitive and personal data, *e.g.*, Vehicle Plate Number (VPN) or vehicle IDs, should be secured. Data eavesdropping and exposure should be prevented by data encryption. The system should not reveal any sensitive personal data during issuing the parking permit and the park vehicle phase.
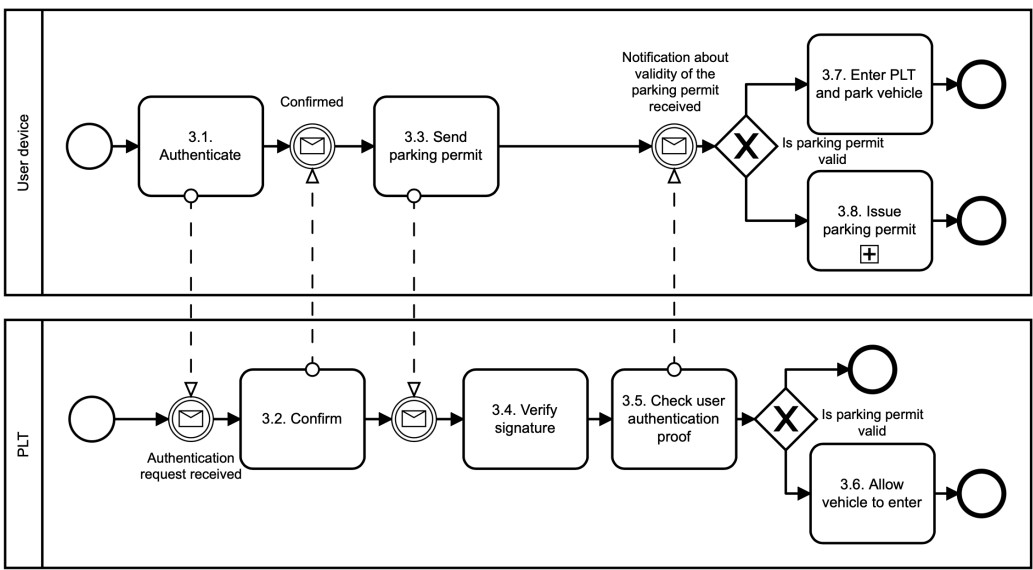

**Figure 4** Park vehicle.

- **Data authenticity and integrity**: All exchanged data (*e.g.*, parking permits, information about available slots, notifications) should be secured against their tampering by unauthorized parties.

  Furthermore, we identify the following system privacy requirements:

- **Data privacy**: All stored and exchanged data should not be exposed to undesired parties and eavesdroppers, *e.g.*, user's vehicle ID, user parking history, and user profiles.
- **Pseudonymity**: A user should be pseudonymous and should be identifiable only in case of certain conditions by PSP. Users should not be identifiable while using the parking system by external and internal parties (PLTs) or other users.
- **Unlinkability**: PSP, PLTs, and other users should not be able to link together the parking actions of the same user (vehicle). The system should not scan VPNs.
- **Conditional traceability**: PSP should not be able to trace users' credentials and their parking actions if the users are honest. PSP should be able to open a user's identity from the parking permit only in case of serious fraud and by cooperation with PLT.
- **Revocation**: PSP should be able to conditionally open the parking permit credentials and identify the user. In a serious incident, PSP can remove a user from the system or remove the user's anonymity. To do so, PSP should collaborate with PLT or, where appropriate, with other trusted third parties.

For data processing, it is necessary that the parking transaction records produced during the system use are stored and processed in a privacy-preserving manner at the PSP under the control of the PLT, thus leading to the following additional security and privacy requirements:

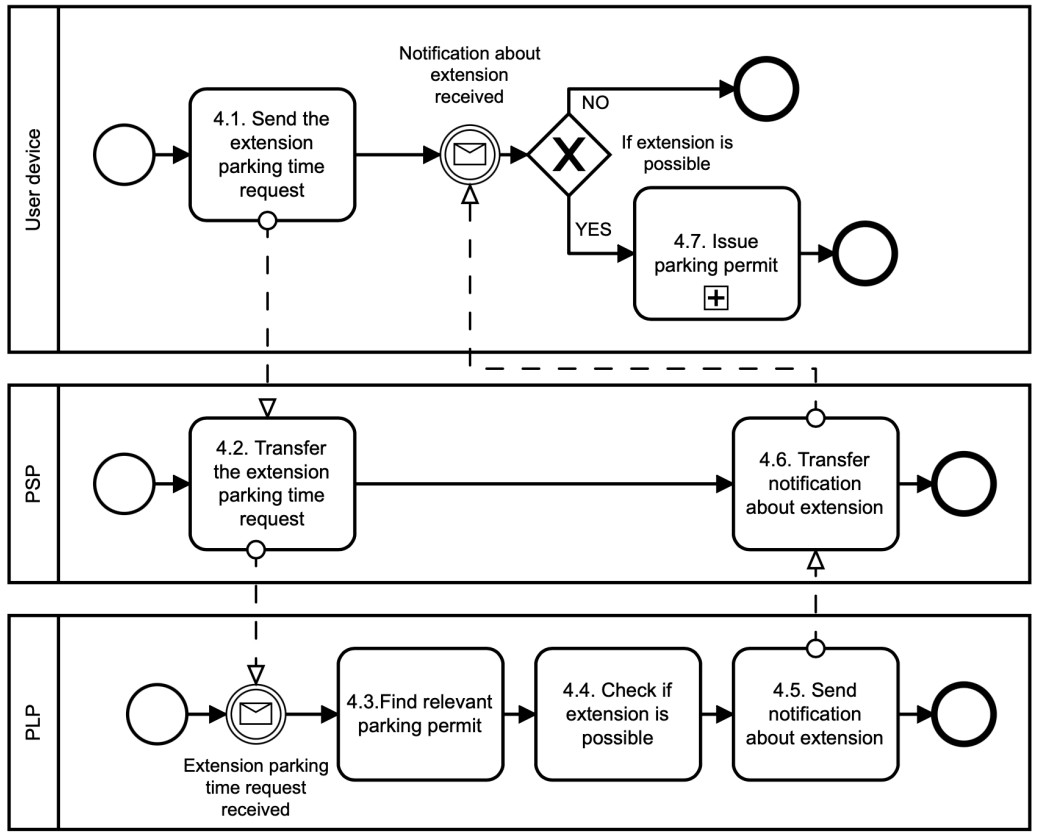

**Figure 5   Extend parking time.**

- **Data minimisation**: The transaction data items stored should be reduced only to the necessary data items for service usage analysis.
- **Index and document privacy**: The encrypted data used for statistics extraction should not reveal any sensitive information about the plaintext data and keywords (used for statistics purpose), to any unauthorized entities including the storing PSP.
- **Query privacy**: The type of statistics being performed should remain confidential, to the storing PSP.
- **Access pattern privacy**: No additional information should be revealed from the search results about the data involved.
- **Query authorization**: Statistics extraction should be limited to authorized entities and authorized keywords only.

## LEGAL ISSUES

The objective of this section is to present the different legal instruments relevant for the deployment of such a service in order to answer the first research question, *i.e.*, **RQ1:** What are the legal instruments, issues, and requirements for the deployment of such a system?; After an explanation of the legal framework surrounding user

[1]The reader should bear in mind that the following lines are not intended to provide a detailed analysis of the application of the EU General Data protection Regulation (GDPR) in the scenario presented.; Regulation (EU) 2016/679 of the European Parliament and of the Council on the protection of natural persons with regard to the processing of personal data and on the free movement of such data, and repealing Directive 95/46/EC, 27 April 2016, OJ L119/1. Far from being exhaustive, we draw attention to the existence of EDPB guidelines on connected vehicles (*EDPD, 2020*). In this article, we will not focus on the Directive 2009/136 and the E-privacy proposal. It should be noted that a connected vehicle might be interpreted as a terminal equipment under the EDPB guidelines on connected vehicles (Guidelines 01/2020). This means that the E-privacy Proposal could then be applied when it is necessary to access the information stored in the vehicle (*e.g.*, when presenting the parking permit to the Parking Lot Terminal(PLT). Moreover, depending on the purpose, consent may or may not be required in the sense of the E-privacy proposal.

identification,[1] the second section focuses on the security requirements in the scenario presented. The legal instruments studied relate to (i) use and deployment of Intelligent Transport Systems (ITS) (*EU, 2010*) (ii) consumer protection (*EU, 2019a*; *EU, 2019b*), and (iii) safety requirement for market placement of vehicles and their components (*EU, 2019c*; *UNECE, 2020*).

## Smart parking scenario and data protection requirements

The GDPR applies to any processing of personal data (*EU, 2016*; Art. 4). In its guidelines on connected vehicles, the European Data Protection Board (EDPB) states that: "Even if data collected by a connected car are not directly linked to a name, but to technical aspects and features of the vehicle, it will concern the driver or the passengers of the car" (*EDPB, 2020*; page 5). Under GDPR, personal data is therefore a broad notion (*Purtova, 2018*). Thus, in the scenario studied, there will be different ways of identifying data subjects, in particular through vehicle identification and payment service. Hence, use of such information fall under the material scope of GDPR.

In the scenario studied, additionally to the requirements of lawfulness (*EU, 2016*; Art. 5), of transparency (*EU, 2016*; Art. 12), of data accuracy (*EU, 2016*; Art. 5.1, d) and the data storage limitation (*EU, 2016*; Art. 5.1, e) (*Knockaert et al., 2021*; Art. 29 Working Party, Opinion 8/2014 on the Recent Developments on the Internet of Things, 16.09.2017, WP 223), a fundamental question is the appropriateness of identifying the service user. Indeed, the EU places at the heart of personal data protection the principle of protection by default and by design (*EU, 2016*; Art. 25, *EDPD, 2019*). To be compliant, one prior question is the need for user identification (*EU, 2016*; Art. 5.1, c). This implies determining, at each stage of the development of the service and according to the activities of the PSP and PLT, whether it is necessary to identify the person. Even if an identification is possible by the PSP with personal information such as mobile phone number, bank card and the credential, the minimisation principle is facilitated by the use of a third party, such as IDentity Provider (IDP), to avoid the collection of unnecessary personal data by the PSP. Additionally, each entity (IDP, PSP, and PLT) has no access to the same personal data.

Secondly, if identification is necessary, each entity responsible for the processing must favour the use of pseudonymisation (*EU, 2016*) (Art. 4.5) techniques. Indeed, the user is identified only in some situations by the PSP and the pseudonymisation is favoured for the PLT. The same credential than the one created by the PSP is pseudonymised for the PLT because these privacy-enhancing technique is sufficient to fulfil its purpose. The pseudonymisation is reinforced by unlikability parameters (PSP, PLTs or other users should not be able to link together parking actions of the same user if the parking permit is not recorded for a long period of time by the PLT (principle of storage limitation contained in Art. 5 *EU, 2016*). Finally, it is important to stress that the principle of minimisation is not only about the need or not to identify the data subject, but also about the need to determine an access policy to the personal data processed. Article 25.2 of the GDPR specifies that the control of accessibility to personal data is an integral part of the default data protection principle and states that: "In particular, such measures shall ensure that by default personal data are not made accessible without the individual's intervention to an

indefinite number of natural persons''. In this respect, the parking space reservation service and the payment service should be able to log who accessed to the data subject's data and the possibility to determine whether they have consulted or modified the information are two security measures that should be implemented (*Dumortier, 2018*).[4]

The EDPB states that: "the plurality of functionalities, services and interfaces (*e.g.*, web, USB, RFID, Wi-Fi) offered by connected vehicles increases the attack surface and thus the number of potential vulnerabilities through which personal data could be compromised (⋯) In addition, personal data stored on vehicles and/or at external locations (*e.g.*, in cloud computing infrastructures) may not be adequately secured against unauthorized access'' (*EDPB, 2020*), pages 11–12.[5] According to article 32 of the GDPR, data controllers and data processors have to implement appropriate technical and organizational measures to ensure security of personal data, considering the state of the art, the costs of implementation, the type of personal data, and the potential risks. Several security measures are planned, notably for the communication between users and the PSP and the communication between the PSP and the PLT (see also Section 'Scenario Phases' concerning the architecture and the protocols used and Section 'Privacy-Preserving Parking Solution' regarding the cryptographic design). In particular, the cryptography and the pseudonymization technologies are cited by the GDPR as security measures making identification of data subjects more complex.

## Additional privacy and security requirements
### Deployment and use of intelligent transport systems

In Directive 2010/1024 on the framework for the deployment of ITS in the field of road transport, ITS are defined as "systems in which information and communication technologies are applied in the field of road transport (⋯) and in traffic management and mobility management (⋯)" (*EU, 2010*; Art. 4.4). The provision of services for (i) information on parking places and (ii) reservation of parking places for trucks and commercial vehicles are two priority actions for the Directive (*EU, 2010*; Art. 3). Thus, if the smart parking service also targets commercial vehicles, the PSP might be qualified as provider of an ITS service for reservation of parking places. PSP and/or PLT may also be considered as providers of an ITS information service on parking places. Information on availability may indeed constitutes a preliminary step for reservation of a parking place.

ITS directive contains specific privacy and security requirements where personal data are processed for the operation of an ITS application or service (*EU, 2010*; Art. 10). First, data processing must be pursued in compliance with the GDPR. Second, personal data can be processed only where necessary for the performance of the application/service. Use of anonymous data is strongly suggested. Third, integrity and confidentiality of the data must be ensured. As explained above, personal data is indeed processed. Nevertheless, the data seems necessary in relation to the provided service and in accordance with a data minimization perspective.

### Consumer protection rules

The definition of user used for the smart parking scenario is sufficiently broad to include consumer protection law. Indeed, consumer protection rules may apply if the user of the

smart parking service is a natural person acting outside its professional activity (*i.e.,* in Business-to-Consumer (B2C) relationships). In this context are especially relevant, (i) the Directive 2019/770 on contracts for supply of digital contents and services and (ii) the Directive 2019/771 on sale contracts of goods. The first Directive applies to conformity assessment of digital contents/services while the second applies to conformity assessment of good incorporating or interconnected with digital contents/services (*EU, 2019b*; *Sein, 2020*). Directive 2019/771 also applies to digital content or services incorporated or interconnected to goods—and which are essential for the performance of the goods—provided under the sale contract of these goods.

In the case at hand, due to the two possibilities for users to interact with the parking system, both Directives may apply depending on the means used in order to initiate the parking permit request. When the reservation process is enabled with a standalone application available on the user's mobile phone, the PSP who serves as a software interface between the user and the PLT could be qualified as digital content or service provider under Directive 2019/770 (*EU, 2019a*; recital 19). On the contrary, if the parking permit request process is triggered by a dedicated on-board unit, this device will meet the definition of good with digital elements according to Directive 2019/771 (*Carvalho, 2019*). Both directives highlight importance of security updates and requires that such updates are provided to the consumer in order to keep the good or digital contents/services conform (*EU, 2019a*; Art. 8., *EU, 2019b*; Art. 7.3, *Beale, 2021*). Both provisions highlight that provider of digital contents/services or sellers of goods will not be liable for lack of conformity if the consumer chooses not to install update, only if they have been informed of the importance of the updates to maintain conformity.

### Vehicle safety requirements

In order to obtain EU type approval (*i.e.,* homologation), manufacturers of vehicles, vehicles systems and components must comply with Regulation 2019/2144 (hereafter the "Vehicles General Safety Regulation", *Batura et al., 2021*). Manufacturers must demonstrate compliance with several technical regulations adopted by the EU or the United Nations Economic Commission for Europe (UNECE), including on protection against cyberattacks (*EU, 2019c*; Art 4.5 d). Even if the vehicles general safety Regulation applies primarily to vehicles manufacturer, it may still apply to the scenario studied depending on choices made for the specific architecture of the parking permit request process and the means used to initiate this process (*e.g.,* if the reservation process is made through an on-board unit developed partly or wholly by the vehicle manufacturer).

*Focus on UNECE Regulation n155 on vehicles cybersecurity.* Through homologation, EU law imposes compliance with UNECE Regulation n 155 (*EC, 2021*), which aims to ensure protection of vehicles and their functions against cyber threats to their electrical and electronic components (see Art. 2.2). This text requires that vehicles manufacturers have a CyberSecurity Management System (CSMS). This CSMS must go through a certification process and applies to the entire lifecycle of the vehicle types for which homologation is sought. Under this Regulation the notion of "Vehicle Type" designates vehicles that do not

[6]To comply with this requirement, vehicle manufacturers have to demonstrate the possibility to identify and manage cyber risks linked to their supply chains. This means, among others, being able to (i) identify risks associated to components or services of suppliers and (ii) manage the risks associated to providers of connected services on which vehicles may rely. To that extent, UNECE considers this requirement as implying implementation of information sharing process on cyber risks with suppliers and joint process of incident management. Use of contractual agreement defining cyber security requirements is heavily recommended. Hence, this requirement produces effects on suppliers as it creates a duty to collaborate with the vehicle manufacturers (*UNECE, 2021*).

present differences for essential features of their electrical/electronic and external interface architecture. To this end, the Regulation contains requirements concerning the CSMS in general (*i.e.,* independently from of the manufacturers' vehicles types) and requirements directed toward each vehicle type (*Goldstein, 2020*) (also see Articles 7.3 to 7.3.6 of *UNECE (2021)* for the requirements directed toward vehicle types). Only requirements relating to the CSMS of the manufacturer are presented below. Nevertheless, we highlight that, this Regulation imposes application of a risk identification process for each vehicle type. To that extent, critical elements of vehicles such as the one ensuring connectivity and the parts of the architecture enabling data exchange must be identified (*UNECE, 2021*). The following lines explain the potential application of this regulation within the context of this article.

*Cybersecurity management system requirements.* In order to certify a CSMS, the approval authorities must verify that different processes are implemented by the vehicle manufacturer (*UNECE, 2021*; Art. 7.2) and in order to identify the risks, threats and vulnerability to which vehicles of the manufacturer are exposed. An annex to the regulation identifies high level threats/vulnerabilities and sub level threats/vulnerabilities (*e.g.,* loss of data within cloud infrastructure, loss of data confidentiality/integrity) that must be covered by the CSMS. As specified by *UNECE (2021)*, risks linked to use of connected services are especially relevant in the process. Another requirement is the implementation of procedures to verify proper management of identified risks. To comply with this requirement, a list of mitigation measures, annexed to the Regulation, that include, among others, use of access control to personal data. Additionally, vehicle manufacturers must demonstrate to certification authorities, how the CSMS handles the dependencies and risks stemming from its supply chain.[6]

*Application of UNECE Regulation n155 to the smart parking service.* Regarding the application of this regulation in the context of this article, different scenarios must be distinguished. First, the parking permit request process can be initiated by the user with an on-board unit integrated in the vehicle and developed by the vehicle manufacturer, *i.e.,* the Original Equipment Manufacturer (OEM). Hence, this unit, as part of the vehicle, will be taken into account by the manufacturer within the assessment for compliance with the UNECE regulation. Second, the on-board unit used to initiate the parking permit request might be developed by another entity (*e.g.,* a tier one or tier two supplier) and integrated in the vehicle by the OEM. In this second scenario, the regulation will create requirements for the OEM (*e.g.,* assessing if the unit is a critical element of the vehicle). It will also apply to the supplier of the device which needs to cooperate with the manufacturer to handle supply chain related cyber risks (*e.g.,* see *Upstream, 2022*; *Upstream, 2021*; *Bittner et al., 2021*). Third, the parking permit request may be enabled with a digital application developed by a third party (*e.g.,* the PSP) in association with the vehicle OEM. This scenario raises the question of the qualification of the PSP in relation to the OEM's supply chain. In this context the European Union Agency for Cybersecurity (ENISA) considers that a software provider can be considered as a tier one provider when having direct contractual relationship with the OEM (*ENISA, 2016*). Thus, if the digital application is developed in

collaboration between the OEM and the application provider, the regulation requirements linked to management of the supply chain related to cyber risks may apply. Consequently, where a relation exists between the OEM and the service provider (through the on-board unit or an application), the PSP shall be able to demonstrate that its security measures (*e.g.*, Section 'Privacy-Preserving Parking Solution') allow for the OEM to comply with the requirements mentioned above.

In a last scenario the mobile application used for the permit request process may be developed by the PSP or a third party at the demand of the PSP without involvement of the OEM. In absence of contractual agreement with the application provider, the application offered to the user might fall outside the scope of UNECE Regulation (*ENISA, 2021*). However, according to ENISA "the UNECE Regulation (···) applies to all Connected and automated mobility stakeholders (including Operators of Intelligent Transport System) who must ensure that their products and services conform to cybersecurity goal" (*ENISA, 2021*). As stated above, the PSP may indeed be qualified of provider of an ITS.

As a preliminary conclusion and first response to RQ1, Table 1 identifies the legal instruments applicable to the scenario and the main data protection and security requirements to build a privacy-preserving system.

## CRYPTOGRAPHIC PRELIMINARIES

We first outline the used notation needed to understand the cryptographic core of our privacy-preserving parking system. Then, we briefly introduce bilinear pairing maps and weak Boneh-Boyen (wBB) signature (*Boneh & Boyen, 2008*) which are used throughout all our cryptographic design. Finally, we review the protocols on which our scheme is based, namely a short group signature (HDMR18) proposed by *Hajny et al. (2018)*, partially blind WI-Schnorr signature proposed by *Abe & Okamoto (2000)*, and searchable symmetric encryption scheme proposed by *Gan et al. (2019)*.

From now on, the symbol ":" means "such that", "$|x|$" is the bitlength of $x$ and "$||$" denotes the concatenation of two binary strings. We write $a \in_R A$ when $a$ is sampled uniformly at random from $A$. A secure hash function is denoted as $\mathcal{H} : \{0,1\}^* \to \{0,1\}^\kappa$, where $\kappa$ is a security parameter. We describe the Proof of Knowledge (PK) and the Signature of Knowledge (SK) protocols using the notation introduced by *Camenisch & Stadler (1997)* (CS). In particular, the protocol for proving the knowledge of discrete logarithm of $c$ with respect to $g$ is denoted as $PK\{\alpha : c = g^\alpha\}$ and the protocol for proving the knowledge of discrete logarithm of $c$ with respect to $g$ and message $m$ is denoted as $SK\{\alpha : c = g^\alpha\}(m)$.

### Bilinear pairing

Let $\mathbb{G}_1$, $\mathbb{G}_2$, and $\mathbb{G}_T$ be cyclic groups of the same prime order $n$, $p \in \mathbb{G}_1$, $q \in \mathbb{G}_2$, and $\mathcal{O}$ is the point at infinity. $\mathbb{G}_1$ and $\mathbb{G}_2$ are additive groups and $\mathbb{G}_T$ is a multiplicative group. By definition $(q, \mathbb{G}_1, \mathbb{G}_2, \mathbb{G}_T, \mathbf{e}, g_1, g_2)$ is a bilinear group if it satisfies all below properties:

- **Bilinearity**: $\forall x, y \in \mathbb{Z}_n, p \in \mathbb{G}_1, q \in \mathbb{G}_2 : \mathbf{e}(p^x, q^y) = \mathbf{e}(p, q)^{xy}$.
- **Non-degeneracy**: $\forall p \neq \mathcal{O} \ \exists q \in \mathbb{G}_2 : e(p, q) \neq 1 \in \mathbb{G}_T$ and $\forall q \neq \mathcal{O} \ \exists p \in \mathbb{G}_1 : e(p, q) \neq 1 \in \mathbb{G}_T$.

**Table 1  Smart parking: privacy and security requirements.**

| GDPR, *EU (2016)* | ITS directive, *EU (2010)* | Consumer protection (directives 2019/770 and 2019/771), *EU (2019a)*, *EU (2019b)* | Vehicle safety regulation (UN-ECE regulation No 155), *EU (2019c)*, *UNECE (2020)* |
|---|---|---|---|
| Data minimization | Data minimization | Provision of security updates | Adoption of Cybersecurity management system and implementation within the organization |
| Pseudonymisation and encryption | Data anonymisation | Information on security updates availability and importance to maintain conformity of goods, contents or services. | Process for identification of risks, threats and vulnerabilities |
| Access control | Data integrity and confidentiality | | Requirement to classify risks, assess risks probability and identify treatment measures (including impact assessment) |
| Data storage | User choice where sensitive personal data are processed (consent requirement) | | Application of mitigation measures (list of mandatory measures annexed) |
| Secure contractually and technically the transfer of personal data | | | Effectivity test during design and production phases |
| Risk assessment and appropriate level of security | | | Continuous update of the risk assessment |
| | | | Processes to detect and react timely and appropriately to attacks/threats/vulnerabilities |
| | | | Forensic data collection requirement |
| | | | Management of supply chain related risks through contractual agreements, information sharing processes and joint incident management |
| | | | Identification of critical elements of vehicles |

- **Computability**: There exists an efficient algorithm $\mathcal{G}(1^{\kappa})$ to compute $\mathbf{e}(p,q)$.

In this work, we consider the case $\mathbb{G}_1 \neq \mathbb{G}_2$ that is when $\mathbf{e}$ is an asymmetric bilinear map and the Decisional Diffie–Hellman (DDH) assumption holds.

### Weak Boneh-Boyen signature

The Weak Boneh-Boyen (wBB) signature scheme is a pairing-based short signature scheme. The scheme is provably secure and it is proven to be existentially unforgeable against a weak (non-adaptive) chosen message attack (*Boneh & Boyen, 2008*). The scheme can be easily combined with the zero-knowledge proofs as shown in *Camenisch, Drijvers & Hajny (2016)*. This makes it possible to prove the authorship of signed messages in an unlinkable and anonymous manner. Below is a brief illustration of the wBB signature (*Boneh & Boyen, 2008*):

- $(pk, sk, syspar) \leftarrow \texttt{KeyGen} \leftarrow (1^\kappa)$: On the input of the security parameter $\kappa$, the algorithm generates system parameters $syspar = (q, \mathbb{G}_1, \mathbb{G}_2, \mathbb{G}_T, \mathbf{e}, g_1 \in \mathbb{G}_1, g_2 \in \mathbb{G}_2)$, computes $pk = g_2^{sk}$, where $sk \in_R \mathbb{Z}_q$, and outputs $sk$ as the private key and $(pk, syspar)$ as the public key.
- $(\sigma) \leftarrow \texttt{Sign} \leftarrow (m, syspar, sk)$: On the input of the message $m \in \mathbb{Z}_q$, the system parameters $syspar$ and the secret key $sk$, the algorithm outputs the signature of the message $\sigma = g_1^{\frac{1}{sk+m}}$.
- $(1/0) \leftarrow \texttt{Verify} \leftarrow (\sigma, m, pk, syspar)$: On the input of the system parameters $syspar$, the public key $pk$, a signature $\sigma$ and a message $m$, the algorithm returns 1 if and only if $\mathbf{e}(\sigma, pk) \cdot \mathbf{e}(\sigma^m, g_2) = \mathbf{e}(g_1, g_2)$ holds, *i.e.*, the signature is valid, or 0, otherwise.

## Short group signature HDMR18

The article (*Hajny et al., 2018*) presents a short and fast group signature scheme (HDMR18) based on the wBB proposal. The signature allows a signer to generate an anonymous signature $\sigma(sk_i, m)$ on a message $m$, where $sk_i$ is the signer's private key. The protocol works as follows:

- $(pk, sk_m, spar) \leftarrow \texttt{Setup} \leftarrow (1^\kappa)$: On the input of the security parameter $\kappa$, the algorithm generates the system parameters $spar = (q, \mathbb{G}_1, \mathbb{G}_2, \mathbb{G}_T, \mathbf{e}, g_1 \in \mathbb{G}_1, g_2 \in \mathbb{G}_2)$ satisfying $|q| = \kappa$. It also generates the manager's private key $sk_m \in_R \mathbb{Z}_q$ and computes the public key $pk = g_2^{sk_m}$. It outputs the $(pk, spar)$ as a public output and the $sk_m$ as the manager's private output.
- $(sk_i, RD) \leftarrow \texttt{KeyGen} \leftarrow (id_i, sk_m)$: On the input of manager's private key $sk_m$ and signer's private identifier $id_i$, the protocol outputs the wBB signature $sk_i = g_1^{\frac{1}{sk_m+id_i}}$ to the signer and updates the manager's revocation database $RD$ by storing $id_i$.
- $\sigma(sk_i, m) \leftarrow \texttt{Sign} \leftarrow (m, id_i, sk_i)$: On the input the signer's private identifier $id_i$, signer's private key $sk_i$, and the message $m$, the algorithm outputs the signature $\sigma(sk_i, m) = (g_1', sk_i', s\bar{k}_i, \pi)$, where:

  - $g_1' = g_1^r$: The generator raised to a randomly chosen randomizer $r \in_R \mathbb{Z}_q$.
  - $sk_i' = sk_i^r$: The signers' private key raised to the randomizer.
  - $s\bar{k}_i = sk_i'^{-id_i}$: The randomized private key raised to the signer identifier.
  - $\pi = SK\{(id_i, r) : s\bar{k}_i = sk_i'^{-id_i} \wedge g_1' = g_1^r\}(m)$: The proof of knowledge of $r$ and $id_i$ signing the message $m$.

- $(0/1) \leftarrow \texttt{Verify} \leftarrow (\sigma(sk_i, m), m, pk, BL)$: On the input of the message $m$, its signature $\sigma(sk_i, m)$, a BlackList ($BL$), and the public key $pk$, the algorithm checks the proof of knowledge signature $\pi$ and checks that the signature is valid with respect to the manager's public key using the equation $\mathbf{e}(s\bar{k}_i \cdot g_1', g_2) \stackrel{?}{=} \mathbf{e}(sk_i', pk)$. The collector also performs the revocation check $sk_i' \stackrel{?}{=} s\bar{k}_i^{id_i}$ for all $id_i$ values stored on the $BL$. If the revocation check equation holds for any value on the blacklist, the signature is rejected. Otherwise, the signature is accepted if all other checks pass.

| **User** | | **Signer** |
|---|---|---|
| (Message Owner) | | |
| $pk = g^{sk}$ | $\mathbb{G}, g, q$ | $sk \in \mathbb{Z}_q$ |

$u, s, d \in_R \mathbb{Z}_q$
$z = (\text{info})$
$a = g^u, b = g^s z^d$

$$\xleftarrow{\quad a,b \quad}$$

$t_1, t_2, t_3, t_4 \in_R \mathbb{Z}_q$
$z = (\text{info})$
$\alpha = a g^{t_1} pk^{t_2}, \ \beta = b g^{t_3} z^{t_4}$
$\varepsilon = (\alpha, \beta, z, m)$
$e = \varepsilon - t_2 - t_4 \bmod q$

$$\xrightarrow{\quad e \quad}$$

$S = e - d \bmod q$
$R = u - S \cdot sk \bmod q$

$$\xleftarrow{\quad R,S,s,d \quad}$$

$\rho = R + t_1 \bmod q$
$\omega = S + t_2 \bmod q$
$\sigma = s + t_3 \bmod q$
$\delta = d + t_4 \bmod q$

**Figure 6** **WI-Schnorr partially blind signature.**

## Partially blind WI-Schnorr signature

A form of digital signature known as a blind signature conceals the message's content from the signer. The resulting blind signature can then be publicly verified against the original (unblinded) message and used as a regular digital signature. This technology is mostly utilized in privacy-enhancing protocols where the message's owner and signer are separate entities. In a partially blind signature, the signer may include common public information in the signature (for example, an expiration date). Therefore, the verifier needs the message, the common information, and the signature in order to verify the signature's authenticity. The WI-Schnorr signature, which is a partially blind signature based on the Schnorr protocol and maintains the Witnesses Indistinguishability (WI), was proposed by *Abe & Okamoto (2000)*. The WI-Schnorr signature is depicted in Fig. 6. It is deemed that both the signer and the user have already agreed upon the public value "info".

## Searchable encryption: outsourced private information retrieval

A privacy enhancing technology that can facilitate privacy-preserving data processing is Searchable Encryption (SE), which enables storing a dataset in an encrypted form, while remaining searchable. This process relieves the service provider of the responsibility to maintain and protect the data from data breaches, as well as unauthorized use within the system.

Structured Encryption (STE) is a searchable encryption variation that provides balance between efficiency, functionality and security (*Gan et al., 2019*; *Kamara, 2015*). Non-interactive STE schemes produce encrypted structures that can be queried using a single

message containing a token, whereas in interactive schemes, queries are performed through an interactive two-party protocol. Searchable Symmetric Encryption (SSE) schemes are a special case of STE that specializes for keyword search. In this setting, a data owner creates a data structure with efficient search support, such as an inverted index. Each document in the dataset is then encrypted, forming the Encrypted DataBase (EDB) and outsourced to an external search service. This enables performing queries on the dataset without revealing information about the dataset or the queries to the search service. The encrypted dataset consists of a list of document identifier and keyword-set pairs. Queries are performed using a search token, generated by the data owner, that allows the server to search through the index. A search query returns the document identifiers that satisfy the query expression. In general, an SSE scheme includes the following main algorithms (*Gan et al., 2019*; *Kamara & Moataz, 2017*):

- $(sk, \Delta) \leftarrow$ Setup $\leftarrow (1^\kappa, DB)$: Using a security parameter $\kappa$ as input and a database $DB$, consisting of a list of document identifiers and keywords, it outputs the secret key $sk$ and an encrypted data structure $\Delta$ that will be outsourced to the data server. This algorithm varies depending on the specific SSE scheme and the data structures they use.
- $(\Gamma) \leftarrow$ Token generation $\leftarrow (sk, q)$: Using the secret key $sk$ and the query $q$, it returns a query token $\Gamma$, to be used during search.
- $(\Phi) \leftarrow$ Search $\leftarrow (sk, q, \Gamma, \Delta)$: Using the secret key $sk$, the query $q$, and a search token $\Gamma$ submitted by the user, the data server performs the search operations on encrypted data structure $\Delta$, returning the matching documents. The algorithm outputs a set of encrypted documents $\Phi$.

Depending on the SSE scheme, query expressiveness varies, supporting single-keyword, conjunctive, disjunctive or boolean queries.

## PRIVACY-PRESERVING PARKING SOLUTION

In this section, we show how to integrate security and privacy features to the vehicle parking scenario introduced in Parking Section 'Scenario Description'. Furthermore, we answer the second research question, *i.e.*, RQ2: How to build a privacy-preserving system which meets the requirements from RQ1? Which Privacy-Enhancing Technology (PET) can be used in order to protect users' privacy during using the system, *i.e.*, reservation of parking slots and parking vehicle actions?

### Detailed description of our algorithms

In this section, we instantiate the algorithms and protocols of the privacy-preserving parking system presented in the previous section using the wBB signature (*Boneh & Boyen, 2008*) and its efficient proof of knowledge (*Camenisch, Drijvers & Hajny, 2016*). Note that the proposal does not provide non-repudiation property of user proofs, and therefore, the PSP has to be trusted. The proposed solution uses randomly generated parking permit secret keys $sk_{Cred}$ in each Issue parking permit phase. The users' private keys $sk_U$ are static, and they are given to users during the Registration phase. On a high level, we let the users obtain a wBB signature $\Lambda$ on their; private secret keys $sk_U$ from the PSP. This

signature represents the user's access credential. The revocation and identification of the malicious user are possible due to using this signature. Then, the user and the PLT together create the parking permit, which includes the randomized signature $\hat{\Lambda}$. To do so, they use a partially blind signature scheme (*Abe & Okamoto, 2000*). The PSP and the PLT learn nothing about the value $\hat{\Lambda}$ during the Parking permit issuing phase. In fact, the value $\hat{\Lambda}$ is blinded. Finally, the user proves the knowledge of signature $\hat{\Lambda}$ anonymously and efficiently using the Schnorr-like zero-knowledge protocol for proving the knowledge of a discrete logarithm (*Camenisch & Stadler, 1997*) during the Parking vehicle phase. For the conversion from the proof of knowledge to the signature, we use the Fiat-Shamir heuristics (*Fiat & Shamir, 1986*). We present the concrete algorithm and protocol instantiations below. To make our protocols easier to follow, we provide several illustrative figures (namely Figs. 7–10) describing our protocols algorithmically and which can be read from top to bottom.

## Setup

$(pk_{PSP}, sk_{PSP}, pk_{PLT}, sk_{PLT}, spar) \leftarrow$ Setup $\leftarrow (1^\kappa)$: The purpose of this algorithm is to generate and set system parameters and cryptographic keys of the system. On the input of security parameter $\kappa$, the algorithm generates the public system parameters $spar$ (implicit input of all other algorithms), the public keys of the PSP and the PLTs shared by all users $pk_{PSP}, pk_{PLT}$ and their private keys $sk_{PSP}, sk_{PLT}$ which remain secret. The algorithm is run within the **Setup phase**, is initiated by the PSP, and runs between the PSP and all enrolled PSPs. The algorithms consists from two sub-algorithms, one run by PSP (called SetupPSP) and one run by PLT (called SetupPLT):

- $(pk_{PSP}, sk_{PSP}, spar) \leftarrow$ SetupPSP $\leftarrow (1^\kappa)$: The algorithm inputs the security parameter $\kappa$ and generates the bilinear group with parameters $spar = (q, \mathbb{G}_1, \mathbb{G}_2, \mathbb{G}_T, \mathbf{e}, g_1, g_2)$ satisfying $|q| = \kappa$. It also generates the PSP's private key $sk_{PSP} \in_R \mathbb{Z}_q$ and computes the public key $pk_{PSP} = g_2^{sk_{PSP}}$. It outputs the $(pk_{PSP}, spar)$ as a public output and the $sk_{PSP}$ as the PSP's private output. The algorithms is run by the PSP.
- $(pk_{PLT}, sk_{PLT}) \leftarrow$ SetupPLT $\leftarrow (spar)$: The algorithm inputs the system parameters $spar$ and generates the PLT's private key $sk_{PLT} \in_R \mathbb{Z}_q$ and computes the public key $pk_{PLT} = g_1^{sk_{PLT}}$. It outputs the $pk_{PLT}$ as a public output and the $sk_{PLT}$ as the PLT's private output.

## Register

$(\Lambda, sk_U, RD) \leftarrow$ Register $\leftarrow (ID, sk_{PSP}, spar)$: The purpose of this protocol is to add a new user to the system. The Register algorithm is presented in full notation in Fig. 7. On the input of the PSP's private key $sk_{PSP}$ and the user's identifier $ID$, the algorithm outputs the user's private key $sk_U$, user's access credential $\Lambda$ and updates the PSP's revocation database $RD$. The algorithm is run within the **Registration phase** as an interactive protocol between the PSP and the user device. The system user is then able to require parking permits and access parking lots. The PSP inputs its private key $sk_{PSP}$ and the user inputs the identity $ID$. If the $ID$ is valid, the protocol generates user's private key $sk_U \in_R \mathbb{Z}_q$ and outputs the wBB signature $\Lambda = g_1^{\frac{1}{sk_U + sk_{PSP}}}$ and the secret key $sk_U$ to the user over a secure channel and updates the PSP's revocation database $RD$ by storing $ID||sk_U$.

**Issue**

$(sk_{Cred}, CRED) \leftarrow \texttt{Issue} \leftarrow (\Lambda, sk_{PLT}, pk_{PLT}, PD, spar)$: The parking permit is issued after the payment is done. The algorithm is run within the **Issue parking permit phase** between the user device and the PLT through the PSP. The Issue algorithm is presented in full notation in Fig. 8. The algorithm inputs the user's access credential $\Lambda$, user's parking data $PD$, the PLT's secret key $sk_{PLT}$, the PLT's public key $pk_{PLT}$ and system parameters $spar$. It outputs the parking permit secret key $sk_{Cred}$ and the parking permit $CRED$ that consists of the following elements $(\rho||\omega||\sigma||\delta||\hat{\Lambda}||\hat{g}||PD)$:

- $\hat{\Lambda}$: The user's access credential raised to a randomly chosen parking permit secret key $sk_{Cred} \in_R Z_q$, i.e., $\hat{\Lambda} = \Lambda^{sk_{Cred}}$.
- $\hat{g_1}$: The generator raised to a randomly chosen parking permit secret key $sk_{Cred} \in_R Z_q$, i.e., $\hat{g_1} = g_1^{sk_{Cred}}$.
- $PD$: The public parking data $PD$. The $PD$ includes the PLT's identifier $\texttt{PLTid}$, parking time period $\texttt{time\_duration}$ and information about extended parking time EPT. To sign data, the PLT uses its secret key $sk_{PLT}$.
- $(\rho||\omega||\sigma||\delta)$: The PLT's signature on the user's partially blinded message, *i.e.*, blinded values $\hat{\Lambda}$ and $\hat{g}$, and the public parking data $PD$. First, the PLT commits to the public data $PD$ by computing commitments $a = g_1^u, b = g_1^s z^d$, where $z = \mathcal{F}(PD)$. Then, the user partially blinds the; message. In particular, the user; blinds the values $\hat{\Lambda}$ and $\hat{g}$ and computes commitments $\alpha = ag_1^{t_1} pk_{PLT}^{t_2}, \beta = bg_1^{t_3} z^{t_4}$ using the PLT's public key $pk_{PLT}$, the PLT's commitments $(a, b)$ and the public data $z = \mathcal{F}(PD)$. The user generates the hash $\epsilon$ on all these data, derives value $e$ from $\epsilon$, and sends it to the PLT. The PLT computes blind signature $(R, S, s, d)$ on value $e$ using its secret key $sk_{PLT}$. Finally, the user unblind the blind signature and obtains the signature $(\rho||\omega||\sigma||\delta)$.

The parking permit includes blinded user's access token $\hat{\Lambda}$, and therefore, it cannot be used for user identification by PSP in this phase.

**Verify**

$(0/1) \leftarrow \texttt{Verify} \leftarrow (sk_U, sk_{Cred}, \Lambda, CRED, pk_{PLT}, pk_{PSP}, spar)$: The parking is anonymous and unlinkable since the parking permit does not include any linkable or personal information. The algorithm is run within the **Parking vehicle phase** between the user device and the PLT. The Verify algorithm is presented in CS notation in Fig. 9. The algorithm inputs the user's secret key $sk_U$, the parking permit secret key $sk_{Cred}$, the user's access credential $\Lambda$, the parking permit $CRED$, the PSP's public key $pk_{PSP}$, the PLT's public key $pk_{PLT}$, and system parameters $spar$. It checks that the signature on parking permit is valid under the PLT's public key using the equation $\varepsilon = \omega + \delta \stackrel{?}{=} \mathcal{H}(g_1^\rho pk_{PLT}^\omega || g_1^\sigma \mathcal{F}(PD)^\delta || \mathcal{F}(PD) || \bar{\Lambda} || \hat{\Lambda} || \hat{g})$. If the signature is valid, the algorithm checks the proof of knowledge $\pi$ and validity of the user's access credential $\Lambda$ with respect to the PSP's public key using the equation $\mathbf{e}(\bar{\Lambda} \cdot \hat{g_1}, g_2) \stackrel{?}{=} \mathbf{e}(\hat{\Lambda}, pk_{PSP})$. If all checks pass, the parking permit is accepted. Otherwise, the parking permit is rejected. The proof of knowledge protocol is run as follow:

- The PLT generates random authentication challenge $c \in_R \mathbb{Z}_q$ and send it to the user.

- The user computes proof of knowledge $\pi$ and sends it to PLT:

$$\rho_{SkCred}, \rho_{SkU} \in_R \mathbb{Z}_q$$

$$t = \hat{\Lambda}^{\rho_{SkU}} g_1^{\rho_{SkCred}}$$

$$e = \mathcal{H}(\hat{g_1}, \hat{\Lambda}, \bar{\Lambda}, t, c)$$

$$s_{SkCred} = \rho_{SkCred} - e \cdot sk_{Cred}$$

$$s_{SkU} = \rho_{SkU} + e \cdot sk_U$$

$$\pi = (e, s_{SkCred}, s_{SkU})$$

- The PLT verifies the proof of knowledge $\pi$:

$$\widehat{t} = (\bar{\Lambda} \cdot \hat{g_1})^e \, \hat{\Lambda}^{s_{SkU}} \cdot g_1^{s_{SkCred}}$$

$$e \stackrel{?}{=} \mathcal{H}(\hat{g_1}, \hat{\Lambda}, \bar{\Lambda}, \widehat{t}, c)$$

**Update**

$(sk_{Cred}, CRED) \leftarrow \mathtt{Update} \leftarrow (sk_U, sk_{Cred}, \Lambda, CRED, pk_{PLT}, pk_{PSP}, sk_{PLT}, \mathtt{EPT}, spar)$: The purpose of this protocol is to extend parking time of a parking permit. The algorithm inputs the user's secret key $sk_U$, the parking permit secret key $sk_{Cred}$, the user's access credential $\Lambda$, the parking permit $CRED$, the PSP's public key $pk_{PSP}$, the PLT's public key $pk_{PLT}$, the PLT's secret key $sk_{PLT}$, the extension parking time EPT, and system parameters $spar$. The algorithm is run in two steps.

1. $(0/1) \leftarrow \mathtt{Verify} \leftarrow (sk_U, sk_{Cred}, \Lambda, CRED, pk_{PLT}, pk_{PSP}, spar)$: The $\mathtt{Verify}$ algorithm is run first. The user specifies the parking permit $CRED$ for extension parking time by sending $\mathtt{PLTid}$ and $\mathtt{PPID}$ information. The PLT finds corresponding parking permit $CRED$ in its database and stars the $\mathtt{Verify}$ algorithm. If verification is successful, then the algorithm continues, ends otherwise.
2. $(sk_{Cred}, CRED) \leftarrow \mathtt{Issue} \leftarrow (\Lambda, sk_{PLT}, pk_{PLT}, PD, spar)$: The $\mathtt{Issue}$ algorithm is run second. The user chooses and sends a new expiration parking time ETP to PLT. Then, the user and the PLT run together $\mathtt{Issue}$ algorithm using the $PD$ from the old user's parking permit $CRED$ and new ETP to create a new extended parking permit $CRED$.

**Revoke**

$(ID) \leftarrow \mathtt{Revoke} \leftarrow (CRED, RD, spar)$: Thanks to this algorithm, the PSP can identify malicious users from the parking permits using the revocation database $RD$. The algorithm is run within the **Revocation phase** by the PSP. The algorithm inputs parking permit $CRED$ and PSP's revocation database $RD$. It checks $\bar{\Lambda} \stackrel{?}{=} \hat{\Lambda}^{-sk_U}$ for all $sk_U$ in $RD$. The $sk_U$ that holds in the equation is linked with the user's identifier $ID$. By providing the $ID$ to an identity provider, the PSP can revoke malicious users' anonymity and identify the users.

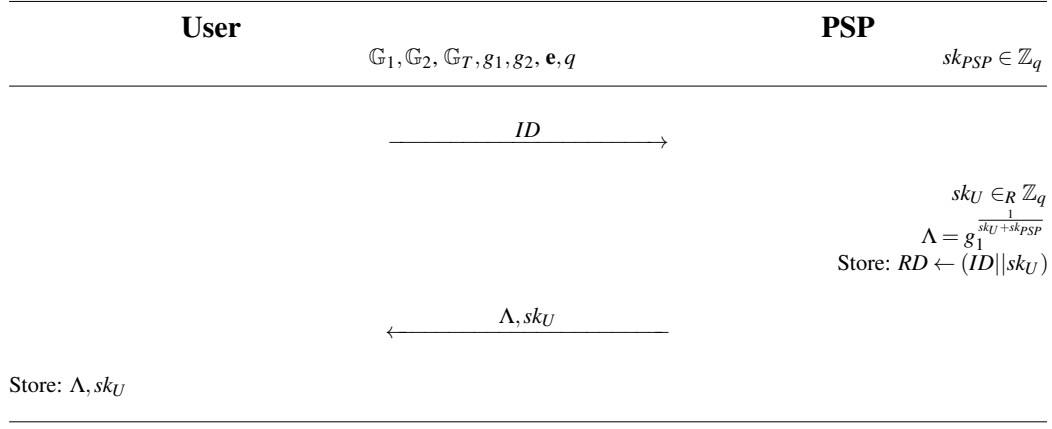

**Figure 7** Register algorithm.

## Optional extension of the system supporting the non-repudiation feature

The proposal of the parking system presented in the 'Description of Our Algorithms' section does not provide non-repudiation features. In fact, the PSP knows all secret keys $sk_U$ of all system users. Thanks to this knowledge, the PSP can revoke users by running the Revoke algorithm. On the other hand, the malicious PSP can forge valid parking permits for all system users, and therefore, falsely accused of committing a crime on anyone in the system. Due to this fact, the PSP must be trusted and honest. However, if the system implementer requires non-repudiation features, we have proposed a solution as well. The solution is based on using a secure two-party computation of wBB signature within the Register algorithm. We refer to *Belenkiy et al. (2009)* and *Ricci et al. (2021)* for more details. Our extension impacts only Registration and Revoke algorithms presented in Section 'Description of Our Algorithms'. The other algorithms remain unchanged. The extended Registration algorithm is depicted in Fig. 10.

The algorithm takes on the input system parameters $spar = (q, \mathbb{G}_1, \mathbb{G}_2, \mathbb{G}_T, \mathbf{e}, g_1, g_2)$ and parameters $(\mathbf{g}, \mathbf{h}, \mathbf{n}, \mathfrak{g}, \mathfrak{h}, \mathfrak{n})$ (*Belenkiy et al., 2009*), where $\mathbf{n}$ is RSA-modulus of size at least $2^{3\kappa}q^2$, $\kappa$ is a security parameter, $\mathbf{h} = \mathbf{n}+1$, $\mathbf{g}$ is an element of the order $\phi(\mathbf{n}) \bmod \mathbf{n}^2$, $\mathfrak{n}$ is RSA modulus such that neither the user nor the PSP knows its factors (*e.g.*, $\mathfrak{n}$ can be provided by a TTP), $\mathfrak{h}$ and $\mathfrak{g}$ are two elements in $\mathbb{Z}_{\mathfrak{n}}^*$ such that $\log_{\mathfrak{g}}\mathfrak{h}$ is unknown and $\mathfrak{g} \in \langle \mathfrak{h} \rangle$. The algorithm is run by the user and the PSP as in main scheme and allows computing user's access credential $\Lambda = g_1^{1/(sk_{PSP}+sk_U)}$ without that the PSP reveals its private key $sk_{PSP}$ and the user its secret key $sk_U$. The algorithm is based on homomorphism of Paillier cryptosystem (*Paillier, 1999*). Fist, the PSP homomorphicly encrypts its secret key $sk_{PSP}$ by computing $e_1 = \mathbf{h}^{\mathbf{n}/2+sk_{PSP}}\mathbf{g}^r \bmod \mathbf{n}^2$ and computes commitment $textgothc_1 = \mathfrak{g}^{sk_m}\mathfrak{h}^{r'} \bmod \mathfrak{n}$. Then, the PSP and the user run the PK protocol:

$$\pi_1 = PK\{(sk_{PSP}, r, r') : e_1/\mathbf{h}^{\mathbf{n}/2} = \mathbf{h}^{sk_{PSP}}\mathbf{g}^r \bmod \mathbf{n}^2$$
$$\wedge \; \mathfrak{c}_1 = \mathfrak{g}^{sk_{PSP}}\mathfrak{h}^{r'} \bmod \mathfrak{n}\}$$

| **User** | **PSP** | **PLT** |
|---|---|---|
| $\Lambda, sk_U, pk_{PLT} = g_1^{sk_{PLT}}$ | $\mathbb{G}_1, \mathbb{G}_2, \mathbb{G}_T, g_1, g_2, \mathbf{e}, q$ | $sk_{PLT} \in \mathbb{Z}_q$ |

$sk_{Cred} \in_R Z_q$
$\hat{\Lambda} = \Lambda^{sk_{Cred}}$
$\bar{\Lambda} = \hat{\Lambda}^{-sk_U}$
$\hat{g}_1 = g_1^{sk_{Cred}}$

$$u, s, d \in_R \mathbb{Z}_q$$
$$PD = \{\texttt{PPID}||\texttt{PLTid}||\texttt{time\_duration}||\texttt{EPT}\}$$
$$z = \mathscr{F}(PD))$$
$$a = g_1^u, b = g_1^s z^d$$

$$\xleftarrow{\quad a, b \quad}$$

$t_1, t_2, t_3, t_4 \in_R \mathbb{Z}_q$
$PD = \{\texttt{PPID}||\texttt{PLTid}||\texttt{time\_duration}||\texttt{EPT}\}$
$z = \mathscr{F}(PD)$
$\alpha = ag_1^{t_1} pk_{PLT}^{t_2}, \beta = bg_1^{t_3} z^{t_4}$
$\varepsilon = \mathscr{H}(\alpha||\beta||z||\bar{\Lambda}||\hat{\Lambda}||\hat{g})$
$e = \varepsilon - t_2 - t_4 \mod q$

$$\xrightarrow{\quad e \quad}$$

$$S = e - d \mod q$$
$$R = u - S \cdot sk_{PLT} \mod q$$

$$\xleftarrow{\quad R, S, s, d \quad}$$

$\rho = R + t_1 \mod q$
$\omega = S + t_2 \mod q$
$\sigma = s + t_3 \mod q$
$\delta = d + t_4 \mod q$

Store: $sk_{Cred}, CRED = (\rho||\omega||\sigma||\delta||\bar{\Lambda}||\hat{\Lambda}||\hat{g}||PD)$

**Figure 8** `Issue algorithm.`

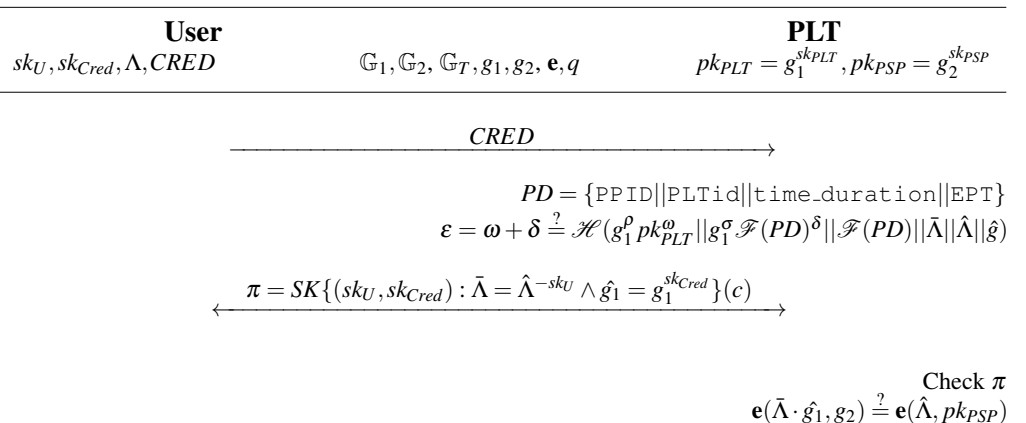

| **User** | | **PLT** |
|---|---|---|
| $sk_U, sk_{Cred}, \Lambda, CRED$ | $\mathbb{G}_1, \mathbb{G}_2, \mathbb{G}_T, g_1, g_2, \mathbf{e}, q$ | $pk_{PLT} = g_1^{sk_{PLT}}, pk_{PSP} = g_2^{sk_{PSP}}$ |

$$\xrightarrow{\quad\quad CRED \quad\quad}$$

$$PD = \{\texttt{PPID}||\texttt{PLTid}||\texttt{time\_duration}||\texttt{EPT}\}$$
$$\varepsilon = \omega + \delta \stackrel{?}{=} \mathscr{H}(g_1^\rho pk_{PLT}^\omega||g_1^\sigma \mathscr{F}(PD)^\delta||\mathscr{F}(PD)||\bar{\Lambda}||\hat{\Lambda}||\hat{g})$$

$$\xleftrightarrow{\quad \pi = SK\{(sk_U, sk_{Cred}) : \bar{\Lambda} = \hat{\Lambda}^{-sk_U} \wedge \hat{g}_1 = g_1^{sk_{Cred}}\}(c) \quad}$$

$$\text{Check } \pi$$
$$\mathbf{e}(\bar{\Lambda} \cdot \hat{g}_1, g_2) \stackrel{?}{=} \mathbf{e}(\hat{\Lambda}, pk_{PSP})$$

**Figure 9** `Verify algorithm.`

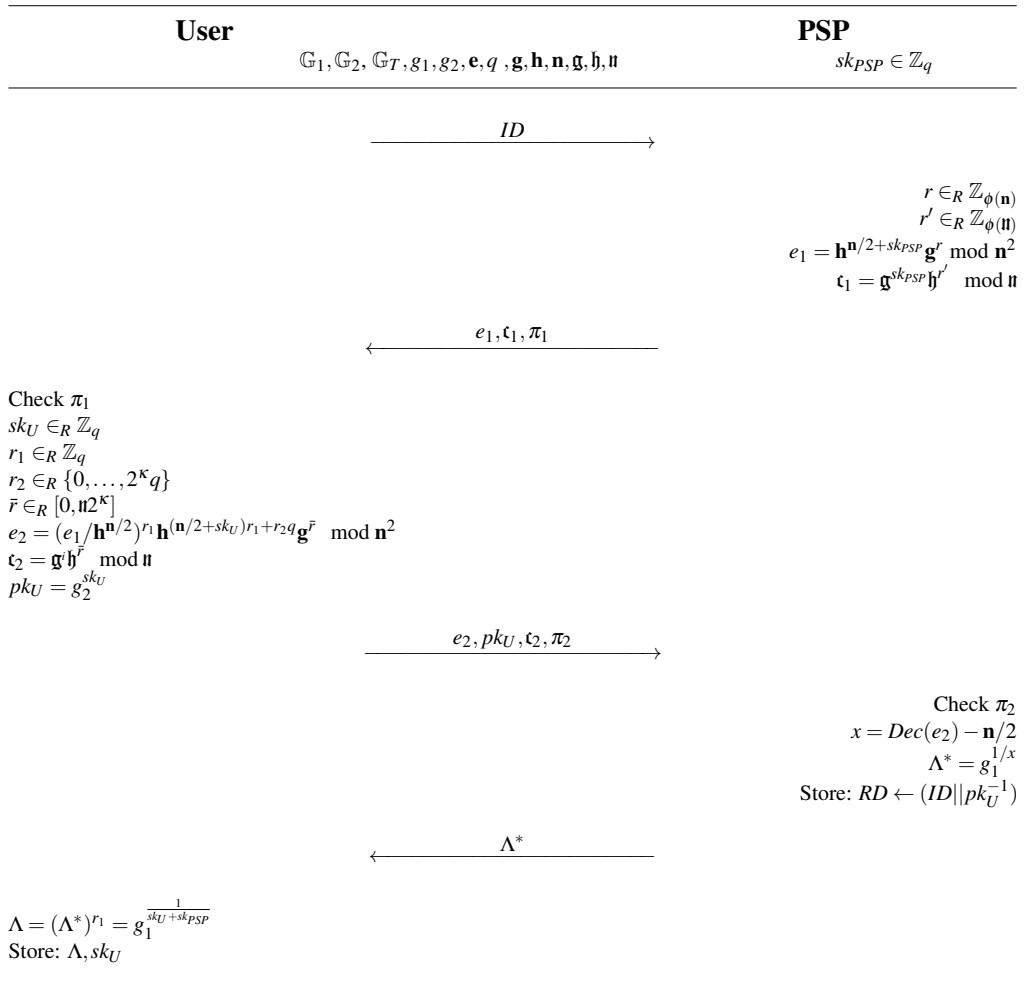

**Figure 10** `Register` **algorithm implementing non-repudiation feature to the parking system.**

If the proof $\pi_1$ is accepted by the user, the user homomorphicly encrypts its secret key $sk_U$ by computing $e_2 = (e_1/\mathbf{h^{n/2}})^{r_1}\mathbf{h^{(n/2+sk_U)r_1+r_2q}}\mathbf{g}^{\bar{r}} \mod \mathbf{n}^2$ and computes commitment $\mathfrak{c}_2 = \mathfrak{g}^{id_i}\mathfrak{h}^{\bar{r}} \mod \mathfrak{n}$ and its public key $pk_U = g_2^{sk_U}$. Then, the PSP and the user run the PK protocol:

$$\pi_2 = PK\{(sk_U, r_1, r_2, sk'_U, u, \bar{r}) : e_2/\mathbf{h^{n/2}} = (e_1/\mathbf{h^{n/2}})^{r_1}\mathbf{h}^{sk'_U}(\mathbf{h}^q)^{r_2}\mathbf{g}^{\bar{r}} \mod \mathbf{n}^2$$

$$\wedge \mathfrak{c}_2 = \mathfrak{g}^{sk_U}\mathfrak{h}^{\bar{r}} \mod \mathfrak{n}$$

$$\wedge 1 = \mathfrak{c}_2^{r_1}(1/\mathfrak{g})^{sk'_U}\mathfrak{h}^u \mod \mathfrak{n}$$

$$\wedge pk_U = g_2^{sk_U}\}.$$

If the proof $\pi_2$ is accepted by the PSP, the PSP decrypts (*i.e.*, Paillier decryption) $x = Dec(e_2) - \mathbf{n}/2$, computes $\Lambda^* = g_1^{1/x}$ and sends it to the user. The user computes $\Lambda = (\Lambda^*)^{r_1}$ and verifies that it is a correct signature on $sk_U$, *i.e.*, $\Lambda = g_1^{\frac{1}{sk_U+sk_{PSP}}}$ holds.

The PSP can open the parking permit *CRED* and track the malicious users by running the modified Revoke algorithm. With the PSP's revocation database *RD* and the parking permit *CRED*, the PSP checks if the equation $\mathbf{e}(\hat{\Lambda}, pk_U^{-1}) \overset{?}{=} \mathbf{e}(\bar{\Lambda}, g_2)$ holds for any of $pk_U$ in its *RD*. If there exists an $pk_U$ for which this equation holds, $pk_U$ is linked with the user's *ID*, which is then sent to the corresponding identity provider to identify the user.

## Security and privacy analysis

The proposed system is built on provable secure cryptographic primitives such as wBB signature (*Boneh & Boyen, 2008*), group signature (*Hajny et al., 2018*), partially blind WI-Schnorr signature (*Abe & Okamoto, 2000*) and secure two-party computation of wBB signature (*Belenkiy et al., 2009*). We refer to these articles for more details on their security analyses. In the security and privacy analysis of our proposal, we adopt the attacker model for the privacy-preserving parking system defined in *Dzurenda et al. (2021)* and apply it to our security and privacy requirements defined in Section 'Parking Scenario Description'. The attacker model considers both internal and external adversaries. In the case of internal attackers, the PSP and the PLT are considered honest-but-curious, while users can act maliciously. Entities omitting the proposed protocols to commit fraud are considered external attackers. Considering this adversary model, we get the security and privacy properties of our system and how they are fulfilled. Note that, we define four lemmas that are in line with our requirements from Section 'Parking Scenario Description'. See Section 'Parking Scenario Description' for more details. Namely, Lemma 5.1 is in line with **Conditional traceability** and **Revocation** requirements, Lemma 5.2 is in line with **Data confidentiality**, **Data privacy**, **Pseudonymity**, and **Unlinkability** requirements, Lemma 5.3 is in line with **Authentication** requirement, and Lemma 5.4 is in line with **Data authenticity integrity** requirements.

**Lemma 5.1** *Revocable anonymity: Users' privacy is preserved as long as they do not try to commit fraud, in which case they can be identified.*

**Proof** During the **Issue parking permit phase**, users provide their unique access credential $\Lambda$ to the PLT through the PSP. This credential is blinded, and therefore, the PLT and the PSP can learn nothing about it. Furthermore, the access credential $\Lambda$ is stored in the parking permit *CRED*, which is presented by the user to the PLT within the **Park vehicle phase**. However, in this case, the users' credentials are randomized, and therefore, mutually unlikable by the PLT. The revocation is possible thanks to $(\bar{\Lambda}||\hat{\Lambda})$ values stored in the parking permit *CRED*. With these two values, the PSP can perform the Revoke algorithm and identify the users. Users traceability by the PSP is possible:

1. **Main scheme (see 'Detailed Description of Our Algorithms')**: The PSP checks $\bar{\Lambda} \overset{?}{=} \hat{\Lambda}^{-sk_{U \in RD}}$ for all $sk_{U \in RD}$ in *RD*. If any $sk_{U \in RD}$ holds in the equation then $sk_{U \in RD} = sk_U$ and $sk_{U \in RD}$ is linked with the user's identifier *ID*:
   $\bar{\Lambda} = \hat{\Lambda}^{-sk_U} \overset{?}{=} \hat{\Lambda}^{-sk_{U \in RD}}$

2. **Extended scheme (see 'Optional Extension of the system supporting the non-repudiation feature')**: The PSP checks $\mathbf{e}(\hat{\Lambda}, pk_{U \in RD}^{-1}) \overset{?}{=} \mathbf{e}(\bar{\Lambda}, g_2)$ for all $pk_{U \in RD}^{-1}$ in

*RD*. If any $pk_{U \in RD}^{-1}$ holds in the equation then the user used corresponding $sk_U$ and $pk_{U \in RD}^{-1}$ is linked with the user's identifier *ID*:

$$\mathbf{e}(\hat{\Lambda}, pk_{U \in RD}^{-1}) = \mathbf{e}(\hat{\Lambda}, g_2^{-sk_U \in RD}) = \mathbf{e}(\hat{\Lambda}, g_2)^{-sk_U \in RD} \stackrel{?}{=} \mathbf{e}(\bar{\Lambda}, g_2) = \mathbf{e}(\hat{\Lambda}^{-sk_U}, g_2) = \mathbf{e}(\hat{\Lambda}, g_2)^{-sk_U}$$

**Lemma 5.2** *Non-traceable and unlinkable reservations: User's actions cannot be bound together by third parties.*

**Proof** The PLT is receiving anonymous blinded user's access credential within the **Issue parking permit phase** and anonymous randomized user's access credential within the **Parking vehicle phase**. No personal or other linkable information is provided during these processes. Due to this fact, the PLT cannot bind any subsequent parking reservation requests of the user. In the same vein, as user's revocable data $(\bar{\Lambda}||\hat{\Lambda})$ are stored in the parking permit *CRED* and only provided to the PSP in case of a fraud attempt, neither the PSP can link user's reservations. The parking permit is always anonymous and unlinkable due to the zero-knowledge property of the proof of knowledge protocol. Distribution of $\hat{\Lambda}$, $\bar{\Lambda}$, $\hat{g_1}$ is random and uniform in $\mathbb{Z}_q$ as $sk_{Cred}$ is selected randomly and uniformly from $\mathbb{Z}_q$:

$$\hat{\Lambda} = \Lambda^{sk_{Cred}}$$
$$\bar{\Lambda} = \hat{\Lambda}^{-sk_U}$$
$$\hat{g_1} = g_1^{sk_{Cred}}$$

**Lemma 5.3** *Fraud avoidance: A user cannot be falsely inquired about not completing a payment process.*

**Proof** The **Issue parking permit phase** is run after the payment for parking is made. The parking permit *CRED* includes the paid parking time, so, any false accusation from the PLT can be denied.

**Lemma 5.4** *Non-repudiation and integrity: Evidences generated from entities interaction can be neither denied nor counterfeited.*

**Proof** The PLT proofs its identity by signing the parking permit *CRED*, the PSP proofs its identity by signing the user's access credential $\Lambda$, and the user proofs the possession of a valid secret keys $sk_U$ and $sk_{Cred}$ within the **Parking vehicle phase**. As a result of the **Issue parking permit phase**, only the user obtains complete parking permit *CRED*, containing the parking reservation details *PD* and revocable data $(\bar{\Lambda}||\hat{\Lambda})$. The PLT gets only partial information about *CRED* (namely *PD* consists of PPID, PLTid, time_duration, EPT). The PLT signs *PD* with blind signature scheme. The signature validity can be verified by everyone, therefore, proofs' integrity is granted.

The signature on a parking permit *CRED* is always accepted if a valid PLT's secret key is used in the signature:

$$\varepsilon = \mathcal{H}(\alpha||\beta||z||\bar{\Lambda}||\hat{\Lambda}||\hat{g}) \stackrel{?}{=} \omega + \delta \stackrel{?}{=} \mathcal{H}(g_1^\rho pk_{PLT}^\omega||g_1^\sigma \mathcal{F}(PD)^\delta||\mathcal{F}(PD)||\bar{\Lambda}||\hat{\Lambda}||\hat{g})$$

$$\alpha = a g_1^{t_1} pk_{PLT}^{t_2} \stackrel{?}{=} g_1^\rho pk_{PLT}^\omega$$

$$= g_1^{(R+t_1)} pk_{PLT}^{(S+t_2)}$$

$$= g_1^{(u-(e-d)\cdot sk_{PLT}+t_1)} pk_{PLT}^{(e-d+t_2)}$$

$$= g_1^u pk_{PLT}^{(-e+d)} g_1^{t_1} pk_{PLT}^{(e-d+t_2)}$$

$$= a g_1^{t_1} pk_{PLT}^{t_2}$$

$$\beta = b g_1^{t_3} z^{t_4} \overset{?}{=} g_1^\sigma \mathcal{F}(PD)^\delta$$

$$= g_1^{(s+t_3)} z^{(d+t_4)}$$

$$= g_1^s z^d g_1^{t_3} z^{t_4}$$

$$= b g_1^{t_3} z^{t_4}$$

$$\varepsilon = e + t_2 + t_4 \overset{?}{=} \omega + \delta$$

$$= S + t_2 + d + t_4$$

$$= e - d + t_2 + d + t_4 = e + t_2 + t_4$$

The signature on randomized user's access credential $\Lambda$ is always accepted if a valid PSP's secret key is used in the signature:

$$\mathbf{e}(\bar{\Lambda} \cdot \hat{g_1}, g_2) \overset{?}{=} \mathbf{e}(\hat{\Lambda}, pk_{PSP})$$

$$\mathbf{e}(\Lambda^{-sk_U \cdot sk_{Cred}} g_1^{sk_{Cred}}, g_2) = \mathbf{e}(\Lambda^{sk_{Cred}}, g_2^{sk_{PSP}})$$

$$\mathbf{e}(g_1^{\frac{-sk_U \cdot sk_{Cred}}{sk_{PSP}+sk_U}} g_1^{sk_{Cred}}, g_2) = \mathbf{e}(\Lambda^{sk_{Cred}}, g_2^{sk_{PSP}})$$

$$\mathbf{e}(g_1^{\frac{sk_{PSP} \cdot sk_{Cred} + sk_U \cdot sk_{Cred} - sk_U \cdot sk_{Cred}}{sk_{PSP}+sk_U}}, g_2) = \mathbf{e}(\Lambda^{sk_{Cred}}, g_2^{sk_{PSP}})$$

$$\mathbf{e}(\Lambda^{sk_{PSP} \cdot sk_{Cred}}, g_2) = \mathbf{e}(\Lambda^{sk_{Cred}}, g_2^{sk_{PSP}})$$

$$\mathbf{e}(\Lambda, g_2)^{sk_{PSP} \cdot sk_{Cred}} = \mathbf{e}(\Lambda, g_2)^{sk_{PSP} \cdot sk_{Cred}}$$

The proof $\pi = SK\{(sk_U, sk_{Cred}) : \bar{\Lambda} = \hat{\Lambda}^{-sk_U} \wedge \hat{g_1} = g_1^{sk_{Cred}}\}(c)$ is always accepted if valid user's secret keys $sk_U, sk_{Cred}$ are used in the proof:

$$e = \mathcal{H}(\hat{g_1}, \hat{\Lambda}, \bar{\Lambda}, t, c) \overset{?}{=} \mathcal{H}(\hat{g_1}, \hat{\Lambda}, \bar{\Lambda}, \hat{t}, c)$$

$$t = \hat{\Lambda}^{\rho_{SkU}} g_1^{\rho_{SkCred}} \overset{?}{=} (\bar{\Lambda} \cdot \hat{g_1})^e \hat{\Lambda}^{s_{SkU}} \cdot g_1^{s_{SkCred}} = \hat{t}$$

$$= (\hat{\Lambda}^{-e \cdot sk_U} \cdot g_1^{e \cdot sk_{Cred}}) \hat{\Lambda}^{(\rho_{SkU} + e \cdot sk_U)} \cdot g_1^{(\rho_{SkCred} - e \cdot sk_{Cred})}$$

$$= \hat{\Lambda}^{\rho_{SkU}} \cdot g_1^{\rho_{SkCred}}$$

**Table 2** Benchmark tests of MCL library operations (modular arithmetic and elliptic curve) on Android devices.

| Device: | | OnePlus Nord 5G | | Honor 8X | |
|---|---|---|---|---|---|
| Elliptic curve: | | BN254 [ms] | BLS12_381 [ms] | BN254 [ms] | BLS12_381 [ms] |
| $\mathbb{F}_r$ | addF (addition) | 0.051 | 0.040 | 0.036 | 0.130 |
| | subF (subtraction) | 0.029 | 0.023 | 0.035 | 0.032 |
| | mulF (multiplication) | 0.025 | 0.019 | 0.027 | 0.023 |
| | divF (division) | 0.081 | 0.091 | 0.145 | 0.149 |
| | negF (negation) | 0.016 | 0.019 | 0.020 | 0.023 |
| $\mathbb{G}_1$ | add1 (addition) | 0.021 | 0.046 | 0.054 | 0.120 |
| | sub1 (subtraction) | 0.022 | 0.034 | 0.037 | 0.058 |
| | mul1 (multiplication) | 0.537 | 1.056 | 0.576 | 0.115 |
| | dbl1 (doubling) | 0.020 | 0.024 | 0.066 | 0.023 |
| | neg1 (negation) | 0.016 | 0.018 | 0.020 | 0.020 |
| $\mathbb{G}_2$ | add2 (addition) | 0.030 | 0.064 | 0.058 | 0.166 |
| | sub2 (subtraction) | 0.027 | 0.052 | 0.039 | 0.106 |
| | mul2 (multiplication) | 0.397 | 2.135 | 1.196 | 2.597 |
| | dbl2 (doubling) | 0.022 | 0.034 | 0.100 | 0.115 |
| | neg2 (negation) | 0.014 | 0.017 | 0.023 | 0.029 |
| $\mathbb{G}_T$ | powT (power) | 1.545 | 2.843 | 2.070 | 3.667 |
| | mulT (multiplication) | 0.040 | 0.061 | 0.102 | 0.142 |
| | pairT (pairing) | 2.808 | 7.527 | 3.025 | 9.687 |

**Notes.**

$\mathbb{F}_r$ represents finite field $\mathbb{Z}_q$, $\mathbb{G}_1$ is cyclic additive group of order $q$ generated by elliptic curve, $\mathbb{G}_2$ is cyclic additive group of order $q$ generated by elliptic curve, $\mathbb{G}_T$ is the cyclic multiplicative group of order $q$.

## Experimental results

In this section, we provide our experimental results. In particular, we show the efficiency of our proposal on Android devices. We use the Android phones: Honor 8X (chip: Kirin 810, OS: Android 10, RAM: 4 GB) and OnePlus Nord 5G (chip: Snapdragon 765G, OS: Android 11, RAM: 8 GB). In order to perform cryptographic operations, we use the MCL (*Shigeo, 2018*) C++ library (using C++17 version of the ISO/IEC 14882 standard); and Android Native Development Kit (NDK). The Android NDK allows us to execute a program in C/C++ on Android devices instead of using Java libraries, and therefore, to achieve better performance results. The source code of the Android application is available online on the GitLab repository (https://gitlab.com/brno-axe/tacr-crypto/android-mcl-test). Our benchmark test on both phones for different arithmetic operations and fields is presented in Table 2. We measure the time complexity of each MCL operation 10 times and then compute the median from these data. From the table, we can see that the time complexity of operations in $\mathbb{F}_r$ is negligible. They take approximately 30 µs for the BN254 elliptic curve on OnePlus Nord 5G. A similar situation is in the case of operations in $\mathbb{G}_1$ and $\mathbb{G}_2$. The most expensive operations are scalar multiplication mul1, mul2, modular exponentiation powT, and bilinear pairings pairT. These operations have a significant impact on the time complexity of the whole protocol.

**Table 3  Computation complexity of the cryptographic algorithms.**

| Algorithm | User | | PSP | | Total |
|---|---|---|---|---|---|
| | Operations | Time [ms] | Operations | Time [ms] | Time [ms] |
| Register | – | – | 1xmul1, 1xaddF 1xdivF | 0.669 | 0.669 |

| Algorithm | User | | PLT | | Total |
|---|---|---|---|---|---|
| | Operations | Time [ms] | Operations | Time [ms] | Time [ms] |
| Issue | 7xmul1, 4xadd1 2xsubF, 4xaddF | 4.107 | 3xmul1, 1xadd1 2xsubF, 1xmulF | 1.715 | 5.821 |
| Verify | 2xmul1, 1xadd1, 1xaddF, 2xmulF 1xsubF | 1.225 | 3xmul1, 3xadd1 2xpairT | 7.290 | 8.515 |
| Update | 9xmul1, 5xadd1, 5xaddF, 2xmulF 3xsubF | 5.331 | 6xmul1, 4xadd1 2xsubF, 1xmulF, 2xpairT | 9.005 | 14.336 |

To show the complexity of our system, we sum up the algebraic operations used in the cryptographic algorithm (*i.e.*, Register, Issue, Verify, Update and Revoke) for each involved system entity and compute the execution time. To do so, we used data from Table 2. In particular, we consider using the OnePlus Nord 5G Android device and the BN254 elliptic curve. Considering our results in the Table 3, the cryptographic core time complexity is negligible, since it takes ca. 6 ms for Issue, ca. 9 ms for Verify and ca. 14 ms for Update algorithm.

The complexity of the Revoke algorithm is linearly dependent on the number of users in the system. The time complexity of Revoke algorithm for both main scheme (see 'Detailed Description of Our Algorithms') and extended scheme (see 'Optional Extension of the system supporting the non-repudiation feature') based on number of system user is depicted in Fig. 11. The main algorithm requires performance of $N$ operations of mul1, while the extended algorithm requires the performance of $N$ operations of pairT), where $N$ is a number of system users. If we consider OnePlus Nord 5G Android device and 1 million system users, the Revoke algorithm will need ca. 9 min; (*i.e.*, main algorithm) and ca. 47 min; (*i.e.*, extended algorithm) to identify a malicious user. By using more powerful servers and palatalization techniques, the revocation time can be reduced significantly.

## PRIVACY PRESERVING DATA PROCESSING FOR STATISTICS ANALYSIS

During the use of the parking system, transaction data is produced, the processing of which could provide interesting information about the characteristics of the service and possible improvements. This processing, however, needs to be performed in a privacy-preserving way, in order to reap the benefits of this processing without

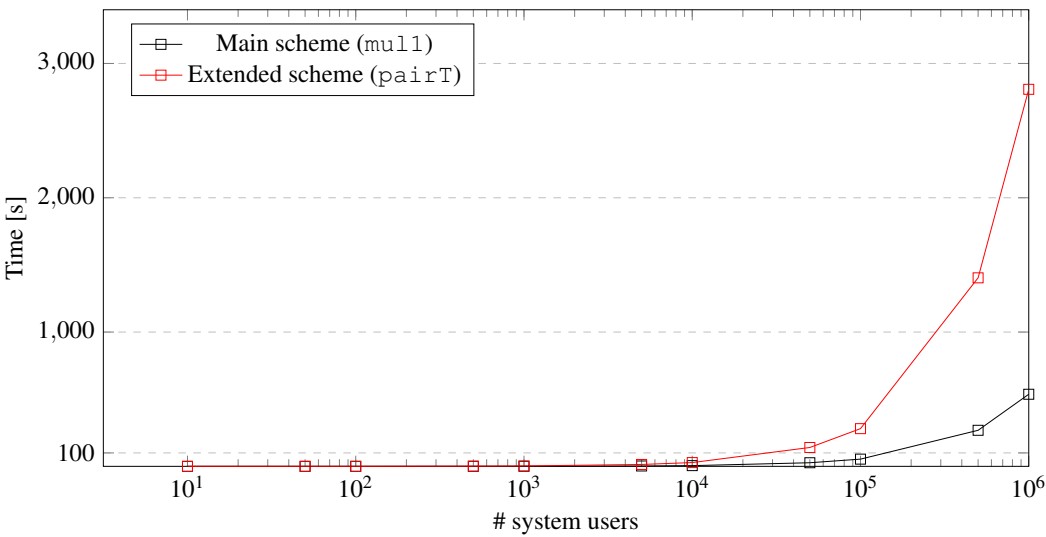

**Figure 11    Time complexity of Revoke algorithm.**

compromising the privacy of the users. To solve this requirement, we need to answer the third research question, *i.e.*, RQ3: How to allow third parties to perform statistical analyses on the parking transaction data, in a privacy preserving way? Which PET can be used to support this task? In the context of the parking scenario, data remain in a clear-text form during the course of a transaction, but once the transaction is completed, the transaction record is moved to long term storage in encrypted form for privacy-preserving statistical analysis. Additionally, following the data minimisation principle, only the necessary data items for analysing the service use is stored in the transaction records. In particular, a parking transaction record includes the following data items:

- **Vehicle classification**: It defines official classification categories used in vehicle licenses.
- **Vehicle type**: It can represent the vehicle power supply, *e.g.*, gas, electric, Liquified Petroleum Gas (LPG).
- **Parking spot type**: It can represent the disability spot, premium spot, short duration spot, long duration spot, secure/closed spot.
- **Services required**: It can indicate washing, charging or any other needs.
- **User affiliation**: It can indicate affiliation with companies/venues, used for special pricing.
- **Parking start timestamp**: Time when parking started and the user has access to the parking lot.
- **Parking end timestamp**: Time when parking has ended and the user must leave the parking lot.
- **Parking transaction cost**: The price that the user must pay for the parking time in the selected parking lot.

Using these data items as search keywords, statistics can be extracted on parking spot demand, peak hours and availability. These statistics can facilitate decisions on pricing strategies and parking spots allocation and management, as well as possible custom offers and packages for specific companies/venues.

The objective of our scheme is to support the following functional and efficiency requirements, additionally to the security and privacy requirements identified in Section 'Privacy and Security Requirements', as they are desired for the SSE scheme and appropriate for the parking scenario:

- **Multi-user functionality**: Searching the dataset is possible for authorized third parties other than the data owner.
- **Query expressiveness** (**boolean query support**): Complex queries need to be supported to enable extracting useful statistics from the dataset.
- **Efficiency**: The search functionality needs to be efficient and scalable, in order for the solution to be applicable in practice.

We propose our MC-SSE scheme (for Multi-Client SSE scheme), which extends the efficient and expressive BIEX SSE scheme (*Kamara & Moataz, 2017*) with multi-user functionality, not supported by the original BIEX scheme. An open source library of the BIEX SSE scheme, known as Clusion,[7] is publicly available, which is particularly attractive in the idea of proposing an extension with experimental results.

## Our MC-SSE system model and overview

The following entities are interacting in the data processing system for the parking scenario, as illustrated in Fig. 12:

- **The data owner** (**D**): Creates an encrypted dataset and outsources it. In our scenario the PLT acts as the data owner.
- **The storage and query server** (**S**): Handles the encrypted dataset storage and performs queries on it.
- **Search clients** (**C**): They are allowed to search on the encrypted dataset. The PSP, other PLTs in the parking system, or any other interested stakeholder can act as a search client.

To achieve the multi-client extension of BIEX, the data owner provides Search clients an authorization token that enables them to create search tokens and submit queries to the Server limited by the keywords contained in the authorization token. With this extension the properties of the BIEX scheme are preserved, *i.e.,* boolean query support with efficient search, while enabling multi-client functionality.

## Refining the security and privacy threat model

As considered in classical SSE schemes, the server S is honest-but-curious, thus fulfilling the search tasks over the database correctly, but attempting to collect as much data as possible. The clients C are malicious and the data owner D is honest. We only consider internal attackers as all the channels between any interacting entities - D-S, D-C and S-C - are authenticated.

[7] Clusion library: https://github.com/encryptedsystems/Clusion

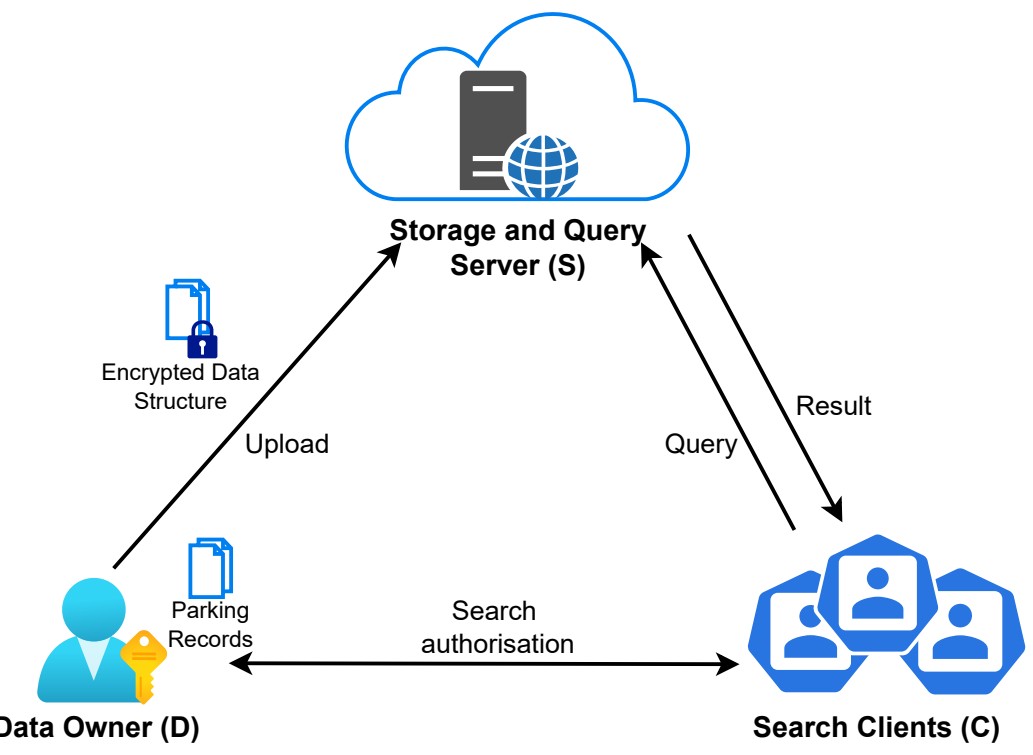

**Figure 12  Multi-client SSE functionality.**

In the light of the system model of section 'Our MC-SSE System Model and Overview', additionally to the security and privacy requirements identified in the 'Privacy and Security Requirements' section—data minimisation, index and document privacy, query privacy, access pattern privacy, the query authorization requirement are revisited for the multi-client SSE scheme as follows:

- **Query authorization**: Only authorized clients on authorized keywords are able to extract statistics from the server.
- **Privilege escalation prevention, as a sublease of the query authorization issue**: Clients must not be able to collude for generating a search token with a superset of combined keywords.

## Our MC-SSE algorithms

The Multi-Client SSE (MC-SSE) scheme consists of the following algorithms:

- $(sk, \text{EDX}, \text{EMM}) \leftarrow$ Setup $\leftarrow (\kappa, DB)$: Taking as input a security parameter $\kappa$ and an index database $DB$, it outputs the secret key $sk$, an encrypted dictionary EDX, and an encrypted multi-map EMM. The algorithm is run by the data owner (D).
- $(\Gamma) \leftarrow$ Authorization token generation $\leftarrow (sk, W_{auth})$: Using as input the secret key $sk$ and a vector of keywords $W_{auth} = (w_1, \ldots, w_q)$, for each keyword $w_i$ in the vector, it creates a sub-token $\Gamma_i = (gtk_i, dtk_i, ltk_j)$ containing a global token $gtk_i$, a dictionary token

$dtk_i$ and for all keywords $w_j$, with $1 \le j \le q$ and $j \ne i$, a local token $ltk_j$. The algorithm is run by the data owner (D) and outputs authorization token $\Gamma = (\Gamma_1, \dots, \Gamma_q)$.

- $(\gamma) \leftarrow$ Search token generation $\leftarrow (\Gamma, \Delta)$: Using as input an authorization token $\Gamma$ and a boolean query $\Delta$, where $\Delta$ is a query written in conjunctive normal form (CNF) as $\Delta_1 \wedge \cdots \wedge \Delta_l$, and where each $\Delta_i = w_{i,1} \vee \cdots \vee w_{i,q}$ is a disjunction, for the first disjunction $\Delta_1$ it calculates the disjunction IEX sub-token $\gamma_1$ as follows:

  - for each keyword $w_{1,j}$ in $\Delta_1$ except the last, it creates the sub-token $\gamma_{1,j} = (gt\bar{k}_{1,j}, dt\bar{k}_{1,j}, lt\bar{k}_{k,j})$ containing the global token $gt\bar{k}_{1,j}$, the dictionary token $dt\bar{k}_{1,j}$ and for all keywords $w_{1,k}$, with $j+1 \le k \le q'$, the local token $lt\bar{k}_{k,j}$. Finally, for the last keyword $w_{1,q'}$ only the global token $gt\bar{k}_{1,q'}$ is kept and the output is the search token $\gamma_1 = (\gamma_{1,1} \cdots \gamma_{1,q'-1}, gt\bar{k}_{1,q'})$.

  For every following disjunction $\Delta_i, 2 \le i \le l$ it computes the sub-token $\gamma_i$ containing the local tokens between every keyword in $\Delta_1$ and every keyword of $\Delta_i$, as follows:

  - for each keyword $w_{1,j}$ in $\Delta_1$ it calculates the vector of local tokens $lt\bar{k}_{j,i} = (lt\bar{k}_{j,i,k}), 1 \le k \le q'$ between the keyword $w_{1,j}$ in the first disjunction and every keyword $k$ in the $i$th disjunction. Then the output is $\gamma_i = (lt\bar{k}_{1,i} \cdots lt\bar{k}_{q',i})$.

  The algorithm is run by the search client (C) and the final output is a search token $\gamma = (\gamma_1, \dots, \gamma_{q'})$.

- $(T)$ Search $\leftarrow$ (EDB, $\gamma$ ): Using as input EDB=(EDX, EMM) and a search token $\gamma = (\gamma_1, \dots, \gamma_{q'})$, for the first search sub-token $\gamma_1 = (\gamma_{1,1} \cdots \gamma_{1,q'-1}, gt\bar{k}_{1,q'})$, the server performs the IEX search as follows:

  - For every element $\gamma_{1,i} = (gt\bar{k}_{1,i}, dt\bar{k}_{1,i}, lt\bar{k}_{k,i}), 1 \le i \le q' - 1$:

    * First it uses $gt\bar{k}_{1,i}$ to query the global multi-map EMM, to recover the set $T_{1,i}$ of document identifiers containing $w_i$.
    * Then it uses $dt\bar{k}_{1,i}$ to query the encrypted dictionary EDX to recover the local multi-maps for $w_i$.
    * Finally, it uses the local tokens $lt\bar{k}_{k,i}$, with $i+1 \le k \le q'$ to query the local multi-maps, to recover the set of document identifiers $T'$ that contain both $w_i$ and $w_k$ and removes them from $T_{1,i}$.

  - For the last element in $\gamma_1, gt\bar{k}_{1,q'}$, it recovers the document identifiers $T_{1,q'}$ containing $w_{q'}$.

  - Finally, the server calculates the set of document identifiers $T_1$, containing the document identifiers $T_{1,i}$ through $T_{1,q'}$.

  For every following search sub-token $\gamma_i = (lt\bar{k}_{1,i} \cdots lt\bar{k}_{q',i}), i \ge 2$, the server:

  - Uses $dt\bar{k}_{1,i}$ from $\gamma_1$ to query the encrypted dictionary EDX to recover the local multi-maps for $w_i$.

  - Then, it uses the local tokens $lt\bar{k}_{k,i}$, with $1 \le k \le q'$ to query the local multi-maps, to recover the set of document identifiers $T_{k,i}$ that contain both $w_i$ and $w_k$.

  - Then it calculates the union of all the common document identifiers $T_i = \bigcup_k T_{k,i}$

– and finally replaces $T_1$ with the intersection of $T_i$ and $T_1$.

At the end, the server outputs the remaining set of identifiers in $T_1$, which is the resulting set of document identifiers for the query $\Delta$. The algorithm is run between the search client (C) and the storage and query server (S).

Note that the Setup and Search algorithms of the MC-SSE scheme are the same as the Setup and Search algorithms of the original BIEX scheme.

## Security and privacy analysis

Our MC-SSE scheme is built over the BIEX scheme for which several SSE requirements have been proven. Based on the MC-SSE threat model presented in 'Refining the Security and Privacy Threat Model' and security and privacy requirements presented in Section 'Privacy and Security Requirements', a security and privacy analysis is conducted below for each of the expected requirements:

**Lemma 6.1** *Index and document privacy: S or any other entities are not able to deduce any sensitive information about the plaintext of the stored encrypted data, nor the associated keywords.*

**Proof** The resulting encrypted data obtained thanks to the MC-SSE Setup algorithm are exactly the same as the ones generated by the BIEX Setup algorithm. As a consequence, our MC-SSE scheme inherits from the requirement Index and document privacy of the BIEX scheme.

**Lemma 6.2** *Access pattern privacy: S is not able to deduce any information about the data from the search results.*

**Proof** The search results obtained thanks to the MC-SSE Search algorithm are exactly the same as the ones obtained from the BIEX Search algorithm. As a consequence, our MC-SSE scheme inherits from the requirement access pattern privacy of the BIEX scheme.

**Lemma 6.3** *Query privacy: S is not able to deduce the type of statistics being performed.*

**Proof** The search method applied by S thanks to the MC-SSE Search algorithm is the exact same method with the BIEX Search algorithm. As a consequence, our MC-SSE scheme inherits from the requirement query pattern privacy of the BIEX scheme. However, although S is unable to deduce the type of performed statistics, it is possible to deduce that a client is doing the exact same request if C is reusing the same authorization token for the exact same request to S. To mitigate that issue, C must be careful not to reuse any elements of the authorization token to S, or should ask for a new authorization token to D (cf. Section 'Our MC-SSE Algorithms').

**Lemma 6.4** *Query authorization: Only authorized clients on authorized keywords only are able to extract statistics from S.*

**Proof** Our original MC-SSE scheme does not prevent itself against any client stealing an authorization token or a search token and issuing an illegitimate request to S. However, the unauthorized usage of tokens can be prevented as follows. D can issue a certificate of

ownership for the authorized client. This certificate signed by D can be computed over a signed randomized accumulator $Acc = g^{\prod_{a_{i,j} \in \Gamma} a_{i,j} \cdot ID_C}$, where $g$ is a group of prime order $q$, $a_{i,j} \in \mathbb{Z}_q$ are the elements of the authorization token $\Gamma = (dtk_i, gtk_i, ltk_j, gtk_q)$, and $ID_C \in \mathbb{Z}_q$ is the *ID* of search client (C). C receiving the certificate is then able to extract his own search token and to adapt the accumulator by removing the elements selected for his search token $\gamma$: $Acc_C = g^{\prod_{a_{i,j} \in \Gamma \setminus \gamma} a_{i,j}}$, where $a_{i,j}$ are all the elements of the authorization token $\Gamma$, excluding the elements of the search token $\gamma$ itself. S can check the validity of the certificate issued for C by computing $Acc_C^{\prod_{a_{i,j} \in \gamma} \cdot ID_C}$ and by checking that the signature is valid with regard to the resulting $Acc_C$. Moreover, the underlying authenticated channel enables S to detect spoofing and replay attacks over the pair - certificate and search token.

**Lemma 6.5** *Privilege escalation prevention: C is not able to collude with other clients to issue a valid search token over a superset of keywords which D did not authorize.*

**Proof** Suppose two clients $C_i$ and $C_j$ with authorization tokens $\Gamma_i$ authorizing keywords in vector $W_{auth_i} = (w_{i1}, \ldots, w_{iq})$ and $\Gamma_j$ authorizing keywords in $W_{auth_j} = (w_{j_1}, \ldots, w_{j_{q'}})$, respectively ($w_{j_k}$ are not elements of $W_{auth_i}$) try to collude to issue a new token for the superset of keywords authorized in $\Gamma_i$ and $\Gamma_j$, to be able to perform cross-searches, *i.e.,* queries combining keywords from the two disjoint authorization tokens. The resulting combined $\Gamma_{ij}$ for the keywords in $W_{auth_{ij}} = (w_{i1}, \ldots, w_{iq}, w_{j_1}, \ldots, w_{j_{q'}})$ will not be usable to perform cross-searches, as although $\Gamma_{ij}$ will contain the global tokens and dictionary tokens of all the combined keywords, it will not contain the local tokens for the combinations of keywords between the two authorization tokens. Therefore, authorization tokens could not be combined to authorize searches on combinations of keywords not already allowed by the initial authorization tokens, and the privilege escalation prevention is supported.

**Lemma 6.6** *Data minimisation: The transaction data items stored are reduced only to the necessary data items for service usage analysis.*

**Proof** D needs to adequately select the set of keywords for limiting keywords to what is necessary for service usage analysis. The selection of keywords is a matter of regulation to respect and a matter of strategy for the company which needs relevant analysis results.

## Experimental results

The MC-SSE evaluation consisted of experiments with up to 1 million documents, containing synthetic parking transactions, resulting in 21 million document-keyword pairs. Experiments were conducted on both the original BIEX scheme and the MC-SSE scheme for the same dataset. The source code of the Clusion BIEX library is available online on the GitHub repository (https://github.com/encryptedsystems/Clusion) and the source code for the multi-client extension of the Clusion BIEX library is available on the on the GitHub repository (https://github.com/atasidou/MC-Clusion). A docker container with the multi-client library extension bundled with a web application for testing its functionality is also available online on the Docker Hub repository (https://hub.docker.com/r/atasidou/multi-client_clusion). Experiments were executed on the Grid'5000 testbed (https://www.grid5000.fr/) with Intel Xeon Gold 6130 (Skylake, 2.10

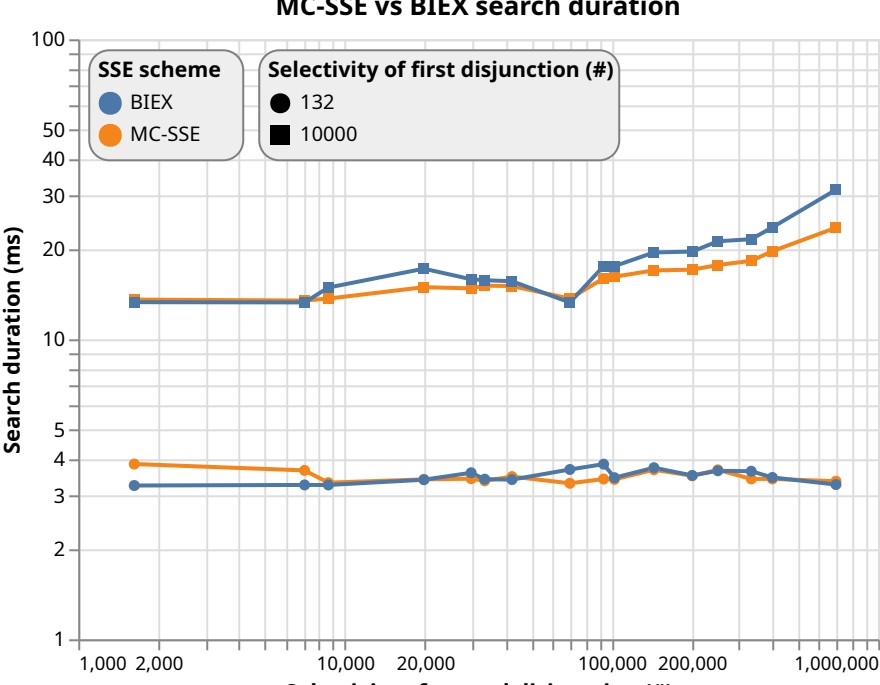

**Figure 13  Boolean search time.**

GHz, 4 CPUs/node, 16 cores/CPU) processors and 60 GB of RAM, running Debian 11 (64-bit) OS. The experimental results confirm the correct functionality of the multi-client extension of the BIEX SSE scheme library. The performance evaluation of the MC-SSE library implementation shows that the properties of the original BIEX SSE scheme algorithm are retained, offering practical and efficient boolean search functionality.

In particular, as illustrated in Fig. 13, the efficiency of the search functionality for MC-SSE is consistent with the original BIEX performance, taking approximately 3–30 ms, to perform a boolean search over 1 million documents (21 million document-id pairs). Note that the query expression includes two disjunctions (sub-queries) with 2 keywords each, of the form $((w \vee x) \wedge (y \vee z))$, with the search time depending on the selectivity of the first disjunction $((w \vee x))$ of the query, *i.e.,* the number of documents returned by the first sub-query. The slightly smaller times in the MC-SSE search duration presented, mainly in higher values of the search times, is due to slight improvements in the Java code for the implementation of the search functions in the Clusion library.

In the multi-client version of the scheme, the main difference is the creation of the authorization token $\Gamma$, which includes a superset of all the sub-tokens for the included keywords, hence being larger in size compared to the equivalent BIEX search token. The experimental evaluation for the overhead introduced by the authorization token $\Gamma$ consisted of creating authorization tokens and BIEX tokens for the same keywords and measuring the creation time and serialized size of the resulting tokens. For both types of

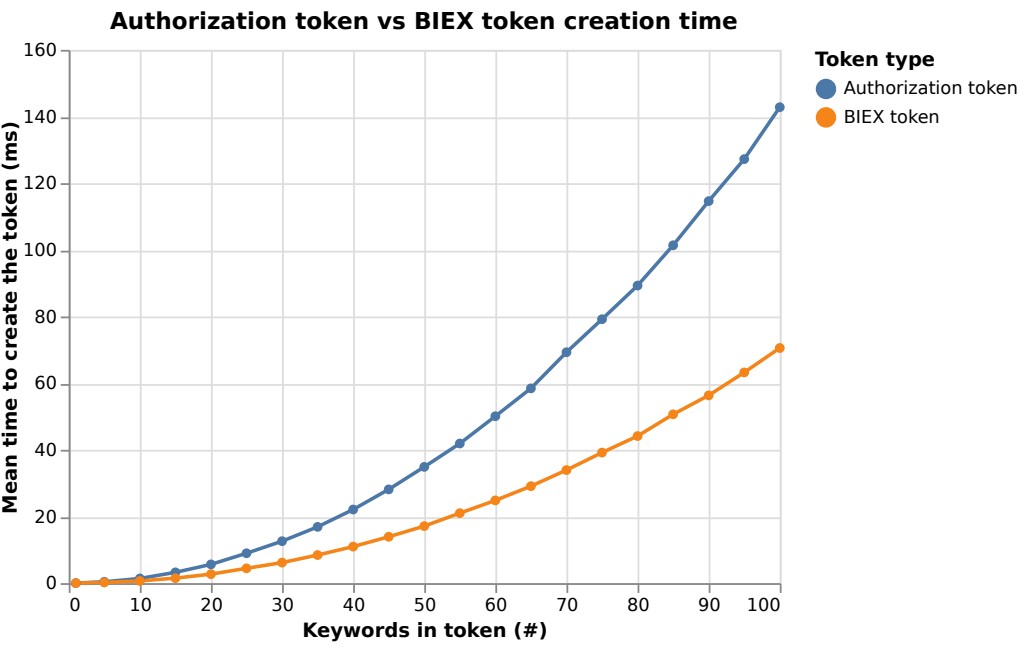

**Figure 14 Token creation time.**

tokens, the creation of tokens for keyword set sizes from $N \in [1, \ldots, 100]$ was evaluated, taking each time the $N$ most frequent keywords in the dataset.

As illustrated in Fig. 14, the creation of the authorization token $\Gamma$ for the multi-client extension of BIEX, shows that the performance of the creation of the authorization token $\Gamma$ displays the same general trend as the BIEX token creation, being exponential with the number of keywords in the token. Despite the creation time of the authorization token $\Gamma$ being higher than the one of the BIEX token, it is just a constant factor higher and remains reasonable in absolute terms. The increased creation time is expected, as the authorization token $\Gamma$ includes approximately double the elements compared to the corresponding BIEX token for the same keyword set, as illustrated in Fig. 15.

## CONCLUDING REMARKS

In this article, we present multidisciplinary work on a comprehensive privacy-preserving system. The work includes research areas starting from regulation compliance analysis, through the design of privacy-preserving parking registration and vehicle parking services to the deployment of privacy-preserving parking data processing features for data analysts. At the beginning of the article, we open up three research questions, namely RQ1, RQ2, and RQ3, which are discussed and addressed in the article.

First, we address the research question RQ1: What are the legal instruments, issues, and requirements for the deployment of such a system? To do so, we provided legal analysis for parking scenarios in compliance with current EU regulations and directives. From a legal point of view, it is obvious that the use of connected objects in cars requires a

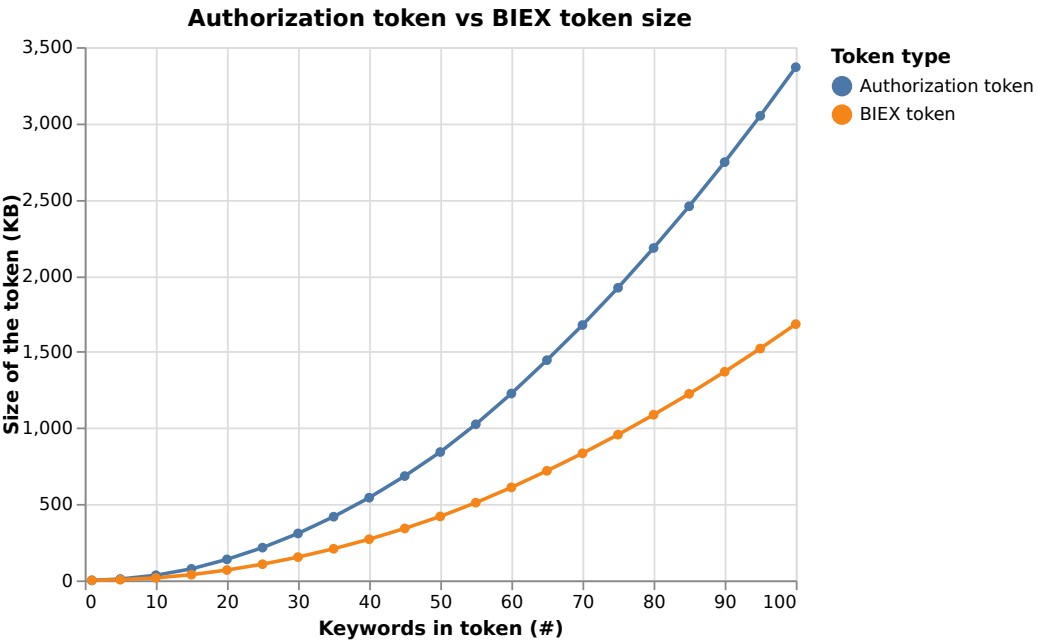

**Figure 15** Token size.

lot of precautions. While the legislation governing the processing of drivers' personal data (GDPR) is a cornerstone, additional security obligations are enshrined in other European legislation. The key principle that drives the most privacy and security requirements is the privacy by design and by default obligation. This approach is accompanied by an appropriate security obligation regarding the risks incurred by users of the smart parking service. This scalable approach is completed by the ITS directive. One has also to keep in mind that, depending on (1) the technical choices made for the implementation of the service and (2) the stakeholder involved, additional sectorial regulations could apply. In particular, if a car manufacturer is engaged in the development of the device/application used to provide the service, application of the EU Regulation on Vehicles General Safety Regulation and UNECE Regulation n155 could be triggered. In such cases, an additional layer of technical and cybersecurity requirements should be met to ensure legal compliance of a smart parking service. Finally, we would like to insist on the fact that an optimal security also requires the implementation of more overall organisational measures.

Second, we addressed the research question RQ2: How to build a privacy-preserving system which meets the requirements from RQ1? Which Privacy-Enhancing Technology (PET) can be used in order to protect users' privacy during using the system, *i.e.*, reservation of parking slots and parking vehicle actions? Here, we used the privacy and security requirements identified from the legal analysis and we propose a novel privacy-preserving parking system. The system protects users' privacy and prevents tracking and profiling of users while using the system (*i.e.*, during parking reservations and vehicle parking actions). On the other hand, the system allows revocation and de-anonymization of malicious users committing fraud. To do so, more system entities, namely PSP, PLT, and

IDP, must collaborate. The cryptographic core of our system is built on provable secure PETs technologies such as group and blind signatures. We provide both security analysis of our system and experimental results.

Finally, we addressed the research question RQ3: How to allow third parties to perform statistical analyses on the parking transaction data, in a privacy preserving way? Which PET can be used to support this task? To do so, we deploy mechanisms for privacy-preserving data processing to our parking system. Completed parking transactions are stored in a dataset, containing only a subset of the data items concerning the transaction information, following the data minimization principle. Using a Searchable Symmetric Encryption (SSE) scheme, this dataset is outsourced to an external search service and stored as a searchable encrypted dataset. The existing efficient and secure BIEX SSE scheme (*Kamara & Moataz, 2017*), with high query expressiveness support was extended to the multi-client setting to allow for authorized parties to perform searches on the encrypted dataset. In this manner, queries can be submitted to the search server to produce statistics on the parking system usage. A security analysis was provided for the proposed solution and experimental results show the applicability and efficiency of the system.

To our knowledge, our work in this article is the first to consider both compliance to regulations (e.g., GDPR) and privacy protection for parking solutions. Most existing solutions mainly focus on some particular technological aspects such as route planning or autonomous parking. Our work is also very comprehensive by presenting both a technical design and an implementation to demonstrate its feasibility. From the figures in the 'Experimental results' section, it is clear that our parking solution is efficient enough to be deployed in practice.

## ACKNOWLEDGEMENTS

The publication only reflects the opinion of the authors. The experiments presented in this article were carried out using the Grid'5000 testbed, supported by a scientific interest group hosted by Inria and including CNRS, RENATER and several Universities as well as other organizations (see https://www.grid5000.fr).

### Funding

This article is supported by European Union's Horizon 2020 research and innovation program under grant agreement No 830892, project SPARTA, and by the Ministry of the Interior of the Czech Republic under grant VJ01030002. Florian Jacques has been supported by the project VIADUCT under reference 7982 funded by Service Public de Wallonie (SPW), Belgium. The funders had no role in study design, data collection and analysis, decision to publish, or preparation of the manuscript.

### Grant Disclosures

The following grant information was disclosed by the authors:

European Union's Horizon 2020 research and innovation program: 830892.
The Ministry of the Interior of the Czech Republic: VJ01030002.
The project VIADUCT under reference 7982 funded by Service Public de Wallonie (SPW), Belgium.
A scientific interest group hosted by Inria and including CNRS, RENATER and several Universities as well as other organizations.

## Competing Interests

The authors declare there are no competing interests.

## Author Contributions

- Petr Dzurenda conceived and designed the experiments, performed the experiments, analyzed the data, performed the computation work, prepared figures and/or tables, authored or reviewed drafts of the article, contributed to the scenario, wrote sections on the cryptographic primitives and privacy-preserving scenario, designed and conducted the evaluation, reviewed the intermediate versions of the article and contributed with concluding remarks, and approved the final draft.
- Florian Jacques conceived and designed the experiments, performed the experiments, analyzed the data, authored or reviewed drafts of the article, contributed to the section on the legal issues, reviewed the intermediate versions of the article and contributed with concluding remarks, and approved the final draft.
- Manon Knockaert conceived and designed the experiments, performed the experiments, analyzed the data, performed the computation work, prepared figures and/or tables, authored or reviewed drafts of the article, contributed to the section on the legal issues, reviewed the article and contributed with concluding remarks, and approved the final draft.
- Maryline Laurent conceived and designed the experiments, performed the experiments, analyzed the data, authored or reviewed drafts of the article, contributed to the scenario, wrote section on the privacy preserving data processing for static analysis, reviewed the intermediate versions of the article and contributed with concluding remarks, and approved the final draft.
- Lukas Malina conceived and designed the experiments, performed the experiments, analyzed the data, authored or reviewed drafts of the article, contributed to the scenario, wrote section on related work, privacy-preserving scenario, reviewed the intermediate versions of the paper and contributed with concluding remarks, and approved the final draft.
- Raimundas Matulevicius conceived and designed the experiments, performed the experiments, analyzed the data, prepared figures and/or tables, authored or reviewed drafts of the article, proposed and discussed scenario, prepared the business process models, managing the paper discussion, reviewed the intermediate versions of the paper and contributed with concluding remarks, submitted and communicated the paper, and approved the final draft.
- Qiang Tang conceived and designed the experiments, performed the experiments, analyzed the data, authored or reviewed drafts of the article, contributed to scenario,

participated in discussions on all paper sections, reviewed the intermediate versions of the paper and contributed with concluding remarks, and approved the final draft.

- Aimilia Tasidou conceived and designed the experiments, performed the experiments, analyzed the data, performed the computation work, prepared figures and/or tables, authored or reviewed drafts of the article, contributed to the scenario, wrote section on the privacy preserving data processing for static analysis, designed and conducted the evaluation, reviewed the intermediate versions of the paper and contributed with concluding remarks, and approved the final draft.

## Data Availability

The raw data is available in the Supplemental Files.

## Supplemental Information

Supplemental information for this article can be found online at http://dx.doi.org/10.7717/peerj-cs.1165#supplemental-information.

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
