# Peer review of "Privacy-preserving solution for vehicle parking services complying with EU legislation"

_PeerJ Computer Science, doi:10.7717/peerj-cs.1165_

## Round 0.1 · original submission · Major Revisions

· Academic Editor

Major Revisions

Enclosed are your reviews. We hope that you will find the reviewers’ comments and suggestions helpful.

Reviewer 1 ·

Basic reporting

The authors present a design of privacy-preserving parking registration and vehicle parking
services system, providing detailed security analysis (as well as experimental results) to support the compliance with regulations such as GDPR, among other related regulations.

The structure of the paper is good and conforms to PeerJ standards. My only complaint is about the way that the citations appear in the text. I think they should be enclosed in parenthesis, otherwise the overall text is hard to read. Please refer to the journal instructions for proper placement. The paper also uses "he" or "she" interchangeably to refer a gender-neutral subject. I recommend using the plural tense for pronouns, and using a gender-neutral word to refer to the subject (such as "the user") throughout the document. I would highlight the header of the tables to make them clearer and more distinguishable for the reader in a quick reading.

English is clear and unambiguous. I've only found minor typos which are commented out as "Additional comments".

The paper is well motivated and presented, and the relevant bibliography is cited. Likewise, the figures are relevant and of good quality. My only concern is when the algorithms in Section 5 are explained. I would advise the authors to explain them algorithmically rather than textually, as they are difficult to follow in current writing.

Regarding the Lemma presentations in Section 5.3, delete "This Lemma is in line with ... ". This text shall be outside the text. When defining the lemma, we need to simply put what we want to prove and the proof.

Experimental design

The research is interesting and original, within the scope of the journal. The research questions are well-defined, relevant, and meaningful. The approach proposed by the authors is fully described and verified, allowing others to replicate it.

Validity of the findings

The novelty of the paper is motivated and convincing. Despite using well-known methods to ensure secure communication, the approach is justified and the overall system security is proven. Moreover, it is justified in accordance with current legal regulations.

Additional comments

- Lines 63-64: Section 3 is cited before Section 2.
- Line 68: "In the last section" -> Section 7
- Line 95: "zero-knowledge proofs *of knowledge*" (remove "of knowledge")
- Line 341: "Parking Service Provider" has already been mentioned
- Line 348: "the user is identified only in some situations _by_ the PSP" (?)
- Line 397: "The first Directive applies*,* to"
- Line 459: Add a reference to "supply chain related cyber risks". There are some recent attacks that you can find in white-reports or other gray literature reports.
- Line 501: "the Decisional Diffie–Hellman (DDH) assumption hold_s."
- Line 516: You mean "or 0, otherwise", right?
- Line 526: Put a comma instead of a period
- Line 528: "On the input*s*"

Reviewer 2 ·

Basic reporting

The paper presents a privacy-preserving system for vehicle parking services. The approach is complying with European Union (EU) legislation, particularly with the GDPR and security requirements defined by current regulations and directives.

The language of the paper is clear and unambiguous, and the diagrams and formalization are well presented.

The Intro presents the research questions faced during the research and the related work presents a nice number of references most of them from 2016 onward, which is perfect (GDPR was released in 2018).
The journal discipline says that for three or fewer authors, list all author names (e.g. Smith, Jones & Johnson, 2004). For four or more, abbreviate with ‘first author’ et al. (e.g. Smith et al., 2005). Please check if there are some of them that need review (e.g. Camenisch et al. (2016))

Although the paper doesn’t follow the PeerJ structure conforms it is presented in this way for clarity. However, I like to find Results and Discussions in a separate section, and some formalizations (e.g. section 4) would be considered an annex.

Figures are well presented but it seems that some of them are not vectorial images and they have escalating issues (12,14 and 15).

Raw data supplied is not necessary.

Experimental design

The research is within the topics of PeerJ.

The paper presents three research questions that clearly identify the paper's purpose. However, would be great to answer the answers specifically again along the paper and remember them in conclusions/discussion.

The research is quite rigorous and performed to a high technical level. I am just missing the C++ standard used.

Methods are described with sufficient detail, and the pipeline could be applied in other fields easily, however, at this point it is not clear how to replicate without adding some implementation noise due to the absence of source code.

Validity of the findings

Impact and novelty are considered assessed, however, it is not clear how commercial applications of parking are working and if they are messing with GDPR, which one would be lightweight for users, or if the battery impact is negligible.


Conclusions are well stated, however, this section needs to remember the RQ proposed and linked with the sections and the results.

Additional comments

The manuscript is written in professional, and unambiguous language. Besides, it also includes mathematical formalization for cryptography involvement.

If there is a weakness, it is in the review of Research questions at the end of the paper, and even a specific discussion section. Besides, the authors should review minor comments presented above.

Of course, would be great for the scientific community to have source code access to algorithms implemented. It also would allow knowing if they are using Off-the-shelf solutions or their implementations.

---

## Round 0.2 · accepted · Accept

· Academic Editor

Accept

The authors have addressed all of the reviewers' comments, and now the paper is ready for publication in PeerJ Computer Science journal.